# CONCEPT BOTTLENECK LARGE LANGUAGE MODELS

**Chung-En Sun, Tuomas Oikarinen, Berk Ustun, Tsui-Wei Weng**

University of California San Diego

`{cesun, toikarinen, berk, lweng}@ucsd.edu`

## ABSTRACT

We introduce Concept Bottleneck Large Language Models (CB-LLMs), a novel framework for building *inherently interpretable* Large Language Models (LLMs). In contrast to traditional black-box LLMs that rely on limited post-hoc interpretations, CB-LLMs integrate intrinsic interpretability directly into the LLMs – allowing accurate explanations with scalability and transparency. We build CB-LLMs for two essential NLP tasks: text classification and text generation. In text classification, CB-LLMs is competitive with, and at times outperforms, traditional black-box models while providing explicit and interpretable reasoning. For the more challenging task of text generation, interpretable neurons in CB-LLMs enable precise concept detection, controlled generation, and safer outputs. The embedded interpretability empowers users to transparently identify harmful content, steer model behavior, and unlearn undesired concepts – significantly enhancing the safety, reliability, and trustworthiness of LLMs, which are critical capabilities notably absent in existing language models. Our code is available at https://github.com/Trustworthy-ML-Lab/CB-LLMs.

## 1 INTRODUCTION

Large Language Models (LLMs) have become pivotal in advancing Natural Language Processing (NLP) tasks. However, their inherent opacity presents significant challenges in ensuring reliability and safety, particularly when their decisions stem from unclear or flawed reasoning. This lack of transparency not only hinders the detection of potential misuse or manipulation, but also makes it difficult to identify and mitigate unsafe outputs. Furthermore, the complexity of their inner workings complicates debugging efforts, making it even more challenging to diagnose and resolve these issues effectively and efficiently.

Several recent works have explored interpretable models in the image domain using concept bottle-necks [5, 10, 14, 16, 19, 20, 21], demonstrating the feasibility of building interpretable models for image classification. In contrast, research on building interpretable models for NLP remains limited. Only a few recent studies [8, 17] have investigated the development of inherently interpretable language models. However, these methods are largely restricted to *text classification* on small datasets and struggle to scale to larger benchmarks or more complex tasks like *text generation*, which is a capability that has become indispensable with the growing dominance of autoregressive LLMs across diverse applications.

Given the limited research on interpretable LLMs, our study is driven by two key goals. First, we aim to enhance the interpretability of LLMs in *text classification* setting by improving scalability for larger benchmarks, reducing development costs, and boosting both model performance and interpretability. Second, we tackle the more challenging *text generation* setting of developing a generative LLM with intrinsic interpretability – an area that is largely unexplored, as existing research focus solely on text classification tasks [8, 17]. By embedding interpretability directly into the LLMs, we enable greater transparency, steering, and control over their behavior. For example, as demonstrated in our case studies, interpretability empowers users to trace the underlying reasoning behind harmful outputs, identifying how inputs and neurons contribute to the generation of unsafe tokens, steering LLMs toward safer response, and forcing LLMs to unlearn undesired concepts. This deeper insight is crucial for enhancing the safety, reliability, and trustworthiness of LLMs.

With the above two goals, in this work, we propose a novel framework for constructing interpretable LLMs that come with intrinsic interpretability. Our method can *transform* any pretrained *black-box LLM* into an *inherently interpretable LLM* by incorporating a human-interpretable Concept Bottleneck Layer alongside a linear prediction layer. We named our method Concept Bottleneck Large Language Models (CB-LLMs), which is, to our best knowledge, the first CBM framework that scales to large text classification and text generation tasks. In Section 3, we formally introduce how to train CB-LLMs for the text classification task, and in Section 4, we present a novel approach to train CB-LLMs for the text generation task. Note that due to the very different nature between the classification task and generation task, a careful design for the training pipeline and algorithm of CB-LLMs for each task is required. To avoid confusion, we refer to these models as CB-LLMs (classification) and CB-LLMs (generation) for each task respectively when the context needs to be clear. Our contributions are as follows:

1. We present a novel framework to build interpretable LLMs for text classification and generation tasks: CB-LLMs (classification) and CB-LLMs (generation). Our CB-LLMs encapsulates the best of both worlds: it matches the high performance of black-box models across multiple settings while offering clear interpretability, a feature absent in existing LLMs.

2. In the classification case, our CB-LLMs (classification) match the accuracy of the standard black-box models and achieves a $1.5\times$ higher average rating compared to the existing works on the faithfulness evaluation. This suggests that our CB-LLMs (classification) provide high-quality interpretability without sacrificing performance.

3. In the generation case, our CB-LLMs (generation) match the performance of the standard black-box models. It provides controllable and understandable generation, allowing further interaction between the user and the LLM. We also developed the first inherently interpretable LLM chatbot that can detect toxic queries and provide controllable responses.

## 2 RELATED WORK

**Concept Bottleneck Models (CBM).** CBM [5] introduces a model structure that incorporates a concept bottleneck layer (CBL), where individual neurons are explicitly designed to learn specific, human-interpretable concepts. This CBL is followed by a final linear layer to produce predictions. Because the activation of the interpretable neurons directly and linearly contributes to the final logits, users can easily understand the model's decision-making process and intervene at the bottleneck layer to correct potential errors.

**CBM in Image Classification.** Recently, CBMs have been revisited in the context of image classification tasks [2, 3, 10, 14, 16, 19, 20, 21]. In the seminal work [5], the authors proposed to train CBMs utilizing human-annotated concept labels, which may be expensive to collect in practice. To address this challenge, recent works [20, 21] leverage concept activation vectors [4] or the multi-modal CLIP model [13] to build CBMs efficiently. However, these approaches still require concept labels to obtain Concept Activation Vector (CAV) or need to restrict the backbone to the CLIP image encoder if concept labels are unavailable, which does not fully resolve the limitation. Recognizing this constraint, [10] proposed Label-free CBM to learn CBMs without relying on concept labels by using the interpretability tool CLIP-Dissect [9]. A recent work [16] further proposed VLG-CBM, to ensure the faithfulness of Vision-based CBMs in image classification tasks, and a new metric called Number of Effective Concept (NEC) is also proposed to control the information leakage and ensure fair comparison between CBMs.

Despite the extensive studies of CBMs in the image classification tasks, to the best of our knowledge, there is still no CBM that scales to large NLP benchmarks or text generation tasks. Consequently, our work focuses on learning an efficient, automated, and high-performance CBM for LLMs.

**CBM in Text Classification.** Two recent works studied the CBM structure in text classification settings. [8] introduced Text Bottleneck Models (TBMs), an interpretable text classification framework that trains a linear predictor on the concept labels generated by GPT-4. Their approach does not involve training the CBL before the linear predictor; instead, they utilize the output score from GPT-4 to replace the output from CBL. Another work, [17], proposed $C^3M$, a framework that merges

human-annotated concepts with concepts generated and labeled by ChatGPT to build the CBM based on GPT-2 and BERT backbone.

While both works aimed to construct interpretable language models utilizing the CBM structure, it's notable that TBM necessitates multiple queries to GPT-4 for each text sample, thereby limiting its applicability to only a small subset of text samples (250 samples) in the datasets. On the other hand, $C^3M$ still depends on human-annotated concepts to augment the concept set, making it challenging to scale to large datasets that lack pre-existing concept annotations. Furthermore, neither work studied the autoregressive generation setting, which is a much more interesting setting given the increasing prevalence of chatbots.

In contrast, our CB-LLMs can scale to large classification datasets of over 500,000 samples and does not require using GPT-4 to label the concepts. CB-LLMs provide interpretability without losing performance and achieves the same accuracy as the non-interpretable black-box counterpart. Furthermore, our proposed approach can handle generation tasks, while existing works are limited to simple text classification. More detailed comparisons are shown in Table 1.

Table 1: Comparison between our CB-LLMs and other interpretable language models in terms of scalability, efficiency, accuracy, and interpretability.

| Methods | Scalability | | Efficiency | | Accuracy | Interpretability |
|---|---|---|---|---|---|---|
| | Text generation setting | Large text classification dataset | Concept labeling w/o querying LLMs | Inference new samples w/o querying LLMs | Same accuracy as black-box model | Provide faithful explanations |
| **Ours:** CB-LLMs | **Yes** | **Yes** | **Yes** | **Yes** | **Yes** | **Yes** |
| **Prior works:** TBM [8] | No | No | No | No | No | No |
| $C^3M$ [17] | No | No | No | **Yes** | No | No |

## 3 CB-LLMs FOR TASK CLASSIFICATION

In this section, we develop interpretable language models for text classification. Section 3.1 presents a cost-effective method for transforming pretrained models into interpretable ones. We evaluate its performance in Section 3.2 and showcase its benefits through a case study in Section 3.3.

### 3.1 METHOD

Our proposed method consists of five steps and is illustrated in Figure 1.

**Step 1: Concept Generation.** The first step is to generate a set of concepts related to the downstream task. To automate this process, we leverage ChatGPT [11] as a replacement for the domain experts. For any text classification dataset $\mathcal{D}$ with $n$ classes/labels, we prompt ChatGPT to generate the concept subset $\mathcal{C}_i$ for each class $i$. Then, the concept set $\mathcal{C}$ is the union of $\mathcal{C}_i$, $\mathcal{C} = \bigcup_{i=1}^{n} \mathcal{C}_i$. We defer the details of prompting to Appendix A.8. Note that our proposed prompting style requires only $n$ queries to ChatGPT to obtain the full concept set, which can be done through the web interface provided by OpenAI at zero cost.

**Step 2: Automatic Concept Scoring.** After generating the concept set $\mathcal{C}$, the next step is to obtain the concept labels for a given text sample $x$ in dataset $\mathcal{D}$ for training. Typically, this stage requires involving domain experts and can be time-consuming. TBM [8] and $C^3M$ [17] leveraged ChatGPT or GPT-4 to automate the labeling process, but their method incurs significant costs due to the high number of API calls required (more than 100M API calls for large dataset like DBpedia). To overcome this challenge, we propose an automatic scoring strategy by utilizing sentence embedding models, which can measure the similarity between each concept and any text sample without the need to query LLMs. We name this strategy as Automatic Concept Scoring (ACS).

For any sentence embedding model $\mathcal{E}$ that encodes a text sample into a fixed-size embedding, we calculate the concept scores $S_c(x) \in \mathbb{R}^k$ for text sample $x$ by calculating the following:

$$S_c(x) = [\mathcal{E}(c_1) \cdot \mathcal{E}(x), ..., \mathcal{E}(c_k) \cdot \mathcal{E}(x)]^\top, \tag{1}$$

where $\mathcal{E}(x) \in \mathbb{R}^d$ denotes the text embedding generated by $\mathcal{E}$, $c_j$ is the $j$-th concept in the concept set $\mathcal{C}$, and $k$ is the size of the concept set. Each component of the vector $S_c(x)$ measures the similarity

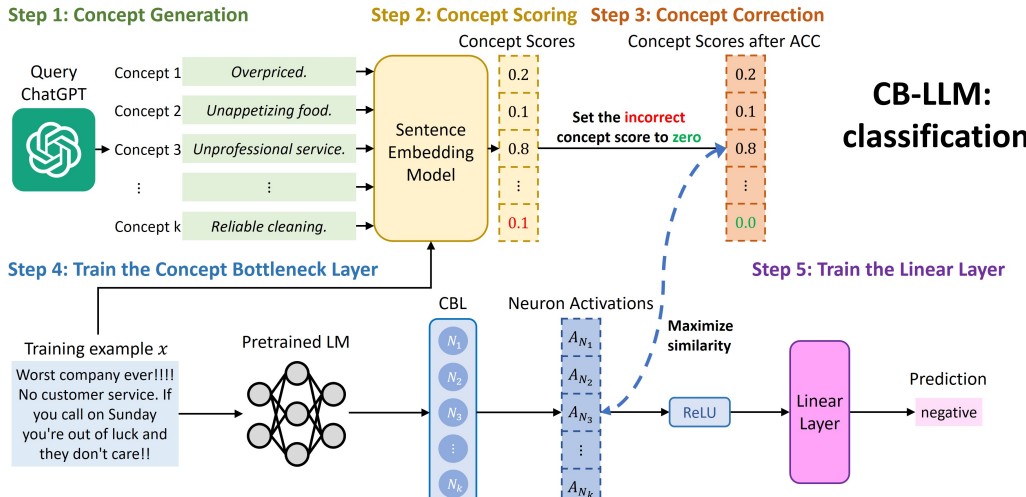

Figure 1: The overview of CB-LLMs (classification). The pipeline consists of five steps: (1) Generate concept set via querying ChatGPT. (2) Automatically label the samples with sentence embedding models. (3) Fix the incorrect concept labels. (4) Train backbone LLM and CBL with the concept labels. (5) Train a linear layer on top of the CBL to make the class predictions.

between the text $x$ and concept $c_j$. These entries represent pseudo concept labels for $x$ that we will use as a learning target for Concept Bottleneck Layer (CBL) in step 4.

We use the off-the-shelf sentence embedding models `all-mpnet-base-v2` from Huggingface [18] for ACS. It achieves better accuracy than labeling by LLMs and requires only a tenth of the time. We will discuss this in Section 3.2.

**Step 3: Automatic Concept Correction (ACC).**  While ACS in step 2 offers an efficient way to provide pseudo concept labels (concept scores), its correctness is dependent on the performance of the sentence embedding model. This may introduce a limitation wherein the concept scores may not align with human reasoning, consequently impacting the learning of the CBL. Notably, this challenge is prevalent in image CBM works that do not rely on human-assigned concept labels [10, 21].

To address this challenge, we proposed Automatic Concept Correction (ACC), a technique to improve the quality of the concept scores generated by ACS in step 2. Recall that in step 1, we generate the concept set $\mathcal{C} = \bigcup_{i=1}^{n} \mathcal{C}_i$ for dataset $\mathcal{D}$ with $n$ classes, where $\mathcal{C}_i$ is the concept subset for class $i$. We define the mapping $\mathcal{M} : c \rightarrow \{1, ..., n\}$ which maps a concept $c \in \mathcal{C}$ to a class: $\mathcal{M}(c) = i$ if $c \in \mathcal{S}_i$. For any text sample $x$ in $\mathcal{D}$, let $y \in \{1, ..., n\}$ be the class label of $x$ and $S_c(x)$ be the concept scores generated by sentence embedding model $\mathcal{E}$ as in Eq.(1). The key idea is to revise $S_c(x)$ to a more accurate concept score $S_c^{\text{ACC}}(x)$ as follows:

$$S_c^{\text{ACC}}(x)_i = \begin{cases} S_c(x)_i, \text{ if } S_c(x)_i > 0, \mathcal{M}(c_i) = y \\ 0, \text{ otherwise} \end{cases} \qquad (2)$$

where $S_c^{\text{ACC}}(x)_i$ is the $i$-th component of vector $S_c^{\text{ACC}}(x)$, and $S_c(x)_i$ is the $i$-th component of vector $S_c(x)$. Intuitively, ACC filters out the negative concept scores and forces every component of $S_c^{\text{ACC}}(x)$ to be zero when the corresponding concept $c_i$ and text sample $x$ belong to different classes. This is achievable because we prompt ChatGPT to generate the concept set for each class separately, thereby providing the mapping $\mathcal{M}$, which associates concepts with their respective classes.

We utilize ACC to correct inaccurate concept scores before training the CBL, leading to a significant improvement in the accuracy of CB-LLMs (classification) ($3.5\%$ in average), which matches those of finetuned black-box models. Further details on the accuracy of CB-LLMs (classification) will be discussed in Section 3.2. Additionally, our ACC strategy does not require any extra queries to ChatGPT and thus requires almost no additional time cost.

**Step 4: Training the Concept Bottleneck Layer (CBL).**  After step 3, we now have the corrected concept scores $S_c^{\text{ACC}}(x)$ for every text example $x$ in dataset $\mathcal{D}$ and are ready for training the Concept Bottleneck Layer (CBL). The goal here is to force the neurons in CBL to activate in correlation with

the pattern of concept scores. We first send the text sample $x$ to a pretrained LM $f_{\text{LM}}$ to get a fixed size embedding $f_{\text{LM}}(x) \in \mathbb{R}^d$. Then, the CBL $f_{\text{CBL}}$ projects the embeddings into a $k$ dimensional interpretable embedding $f_{\text{CBL}}(f_{\text{LM}}(x)) \in \mathbb{R}^k$. To force the $k$ neurons in the CBL learn the concepts, we maximize the similarity between $f_{\text{CBL}}(f_{\text{LM}}(x))$ and $S_c^{\text{ACC}}(x)$ for every $x$:

$$\max_{\theta_1, \theta_2} \frac{1}{|\mathcal{D}|} \sum_{x \in \mathcal{D}} \text{Sim}\big(f_{\text{CBL}}(f_{\text{LM}}(x; \theta_1); \theta_2), S_c^{\text{ACC}}(x)\big), \tag{3}$$

where $\text{Sim} : \mathbb{R}^k \times \mathbb{R}^k \to \mathbb{R}$ can be any similarity function (we adopt *cos cubed* as proposed in [10]), $\theta_1$ and $\theta_2$ are the parameters of $f_{\text{LM}}$ and $f_{\text{CBL}}$, respectively.

**Step 5: Train the linear layer.** After training the CBL, the $k$ neurons in the CBL learn the corresponding $k$ concepts. Let $A_N$ be the neuron activations from CBL, $A_N(x) = f_{\text{CBL}}(f_{\text{LM}}(x))$, we set all the negative activations of $A_N(x)$ to zero through a ReLU function $A_N^+(x) = \text{ReLU}(A_N(x))$. We remove the negative activations as the negation of a concept introduces ambiguity (e.g., it is unclear whether the negative activations imply the absence of a concept or the negation of the semantic meaning of a concept). After obtaining $A_N^+$, we train a final linear layer with sparsity constraint to make the final text classification interpretable:

$$\min_{W,b} \frac{1}{|\mathcal{D}|} \sum_{x,y \in \mathcal{D}} \mathcal{L}_{\text{CE}}(W_F A_N^+(x) + b_F, y) + \lambda R(W_F), \tag{4}$$

where $W_F \in \mathbb{R}^{n \times k}$ is the weight matrix and $b_F \in \mathbb{R}^n$ is the bias vector of the final linear layer, $y$ is the label of $x$, and $R(W) = \alpha \|W\|_1 + (1-\alpha)\frac{1}{2}\|W\|_2^2$ is the elastic-net regularization, which is the combination of $\ell_1$ and $\ell_2$ penalty. $\lambda$ is set to $0.0007$ and $\alpha$ is set to $0.99$.

## 3.2 Experiment

In this section, we evaluate our CB-LLMs (classification) in terms of three crucial aspects: *Accuracy*, *Efficency*, and *Faithfulness*. These aspects are pivotal as our goal is to ensure that CB-LLMs (classification) achieve high accuracy with minimal additional cost while providing human-understandable explanations.

**Setup.** We work with four datasets for text classification: SST2 [15], Yelp Polarity (YelpP) [22], AGnews [22], and DBpedia [6]. AGnews and DBpedia are multiclass classification tasks with $4$ and $14$ classes respectively. YelpP and DBpedia contain $560,000$ training samples which is $2000$ times larger than the largest dataset used in TBM [8] and $20$ times larger than the dataset used in $\text{C}^3\text{M}$ [17]. We generate $208$ concepts for SST2, $248$ concepts for YelpP, $216$ concepts for AGnews, and $476$ concepts for DBpedia. We use `RoBERTa-base` [7] and `GPT2` [12] pretrained model as the backbone for learning CB-LLMs (classification), and compare them with the fine-tuned models (standard black-box models) and the implementation of TBM and $\text{C}^3\text{M}$. Note that TBM and $\text{C}^3\text{M}$ utilize GPT series models for concept labeling, which becomes cost-prohibitive when applied to large datasets. For datasets with $m$ samples and $n$ concepts, this requires $m \times n$ API calls to get all the binary labels (e.g., $560,000 \times 476 = 266,560,000$ API calls for DBpedia). Given the scale of our datasets, their approaches are impractical. Therefore, we opt to use `Llama3-8B-Instruct` [1] as the LLM for labeling and limit the process to $1,000$ samples per dataset to maintain feasibility. We refer to this implementation as TBM&$\text{C}^3\text{M}$.

**Accuracy.** The test accuracy is shown in Table 2. In general, our CB-LLMs (classification) demonstrate high accuracy across various datasets, including large ones such as YelpP and DBpedia. The CB-LLMs implementation without ACC already achieves high accuracy: significantly outperforming TBM&$\text{C}^3\text{M}$ with only a 1~5% gap compared to the standard black-box model. This gap can be further eliminated: it can be seen that our ACC strategy, described in Section 3.1 step 3, improves the accuracy significantly to the level of the standard black-box model. This indicates that ACC can effectively correct inaccurate concept scores and enhance learning on the given task. Overall, our CB-LLMs (classification) sometimes achieve higher accuracy than the standard black-box model (highlighted in blue in Table 2), demonstrating the potential to build interpretable models without the performance trade-offs. Note that our framework is compatible with both encoder-only (e.g. RoBERTa) and decoder-only (e.g. GPT2) backbones. See Appendix A.1 for additional results when using GPT2 as the backbone.

Table 2: Test accuracy of CB-LLMs (classification). CB-LLMs is competitive with the black-box model after applying ACC. Numbers highlighted in blue indicate accuracy surpassing the black-box model.

| Accuracy↑ | SST2 | YelpP | AGnews | DBpedia |
|---|---|---|---|---|
| **Ours:** | | | | |
| CB-LLMs | 0.9012 | 0.9312 | 0.9009 | 0.9831 |
| CB-LLMs w/ ACC | **0.9407** | **0.9806** | **0.9453** | **0.9928** |
| **Baselines:** | | | | |
| TBM&C$^3$M | 0.9270 | 0.9534 | 0.8972 | 0.9843 |
| Roberta-base fine-tuned (black-box) | 0.9462 | 0.9778 | 0.9508 | 0.9917 |

**Efficiency.** CB-LLMs (classification) incur only a small time overhead while achieving interpretability. Our ACS strategy takes about 1.6 hours on the largest YelpP and DBpedia dataset when using `all-mpnet-base-v2` as the sentence embedding model. In contrast, LLM-based labeling, as used by TBM and C$^3$M, takes 8.8 hours to label just 1,000 samples per dataset. The training time of CB-LLMs (classification) is approximately equivalent to the time cost of finetuning the standard black-box model. The detailed comparison of the time cost is shown in Appendix A.2 Table 8.

**Faithfulness.** It is important for an interpretable model to make predictions based on human-understandable and faithful logic. Hence, in this section, we evaluate the faithfulness of CB-LLMs (classification) through human study. Specifically, we design below two tasks for human evaluation:

- **Task 1: Activation Faithfulness.** In this task, workers will be presented with a neuron concept alongside the corresponding top $k$ text samples where this neuron highly activates. Workers need to provide a rating ranging from 1 (strongly disagree) to 5 (strongly agree) based on the agreement observed between the neuron concept and the top $k$ highly activated samples. This task evaluates if the activations of neurons in CBL align with the corresponding concepts they have learned.

- **Task 2: Contribution Faithfulness.** In this task, workers will be presented with explanations from two models for a text sample. Workers need to compare which model's explanations are better. The explanations are generated by showing the top $r$ neuron concepts with the highest contribution to the prediction. Given a text sample $x$, the contribution of a neuron $j$ to class $i$ is defined as $W_{ij}A_N^+(x)_j$, where $W$ is the weight matrix from the final linear layer and $A_N^+$ is the non-negative activations from CBL introduced in Section 3.1 step 5. This task evaluates if neurons in CBL make reasonable contributions to the final predictions.

We conduct human evaluations through Amazon Mechanical Turk (MTurk) for Task 1 and 2 to compare our CB-LLMs (classification) with TBM&C$^3$M. To ensure reliable results, each question in the tasks mentioned above is evaluated three times by different workers. More details about the survey design and interface can be found in Appendix A.7. The results of task 1 (Activation Faithfulness) are shown in Table 3. Our CB-LLMs w/ ACC constantly achieve higher ratings than TBM&C$^3$M. This suggests that the neurons in our CB-LLMs w/ ACC are more interpretable. The results of task 2 (Contribution Faithfulness) are shown in Table 4. Workers consistently express a preference for our CB-LLMs w/ ACC over TBM&C$^3$M. This suggests that the explanations generated by our CB-LLMs w/ ACC are better. We visualize the connection between interpretable neurons and the prediction through the final layer weights, as shown in Appendix A.3. For details on neuron interpretation and the explanations provided by CB-LLMs (classification), refer to Appendix A.5 and A.6.

Table 3: Human evaluation results for Task 1. The higher rating of CB-LLMs (classification) suggests that CB-LLMs are reasonably interpretable to humans.

| Task 1 | Dataset | | | | Average Rating |
|---|---|---|---|---|---|
| Activation Faithfulness ↑ | SST2 | YelpP | AGnews | DBpedia | |
| CB-LLM w/ ACC (**Ours**) | **3.47** | **4.33** | **4.53** | **4.13** | **4.12** |
| TBM&C$^3$M | **3.47** | 2.67 | 2.73 | 2.13 | 2.75 |

Table 4: Human evaluation results for Task 2. Results show that CB-LLMs (classification) provide good explanations.

| Task 2 – Contribution Faithfulness ("which model is better?") | | | | |
|---|---|---|---|---|
| CB-LLMs w/ ACC clearly better | CB-LLMs w/ ACC slightly better | Equally good | TBM&C$^3$M slightly better | TBM&C$^3$M clearly better |
| **27.7**% | **22.3**% | 21.4% | 13.8% | 14.8% |

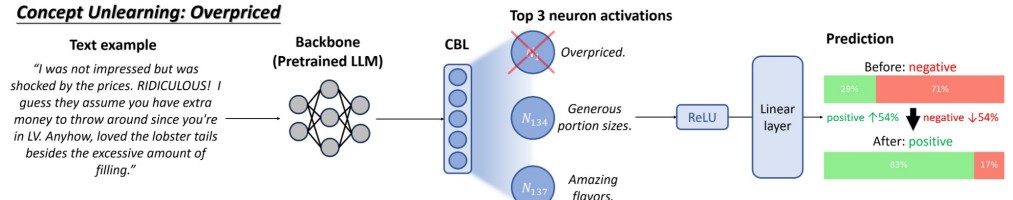

Figure 2: An example of concept unlearning. This example is initially classified as negative due to the customer complaining about the high price, despite the lobster tails being great. After unlearning the concept "Overpriced", the concepts "Amazing flavors" and "Generous portion sizes" dominate the prediction, resulting in a positive prediction.

## 3.3 CASE STUDY ON CONCEPT UNLEARNING

We demonstrate a use case of our CB-LLMs on "concept unlearning," which can enhance prediction fairness by allowing users to remove biased or unfair elements. **Concept Unlearning** involves forcing the model to forget a specific concept, which can be achieved by deactivating a neuron in the CBL or removing its weights from the final linear layer.

Figure 2 illustrates unlearning the concept of "overpriced," which may be subjective or geographically influenced in Yelp reviews (as the standard of overpricing varies across individuals and locations). This adjustment encourages CB-LLMs to focus more on product quality. After unlearning "overpriced," predictions for 2,726 test samples shifted from negative to positive. Subsequently, we employed `bart-large-mnli`, an NLI model, to assess whether these samples indeed contain the concept of "overpriced". Our findings reveal that $2,162$ out of the $2,726$ samples strongly entail "overpriced," accounting for $79\%$. This suggests that most samples with positive predictions were initially classified as negative due to the presence of the "overpriced".

Based on the above case study, we believe our CB-LLMs have great potential to facilitate human intervention such as Concept Unlearning for enhancing fairness, as users can easily remove biased, subjective, or unfair elements that could distort the predictions.

## 4 CB-LLMS FOR TEXT GENERATION

In this section, we investigate a more interesting setting — building interpretable LLMs for generation tasks. In section 4.1, we introduce a novel training design for building controllable and interpretable autoregressive LLMs. In section 4.2, we evaluate the performance of CB-LLMs (generation). In section 4.3, we demonstrate its practical benefits through a case study for toxicity reduction.

## 4.1 METHOD

**Main challenge of the design.** In the generation case, the CBL alone cannot capture all necessary concepts for token prediction, as it is unable to capture all the possible concepts needed for generation. To increase the capability of CBM, a common strategy is to introduce unsupervised (non-interpretable) neurons in parallel with the CBL, as shown in Figure 3 Module 1. This additional unsupervised layer helps provide the necessary broader context for more effective generation. This structure is known as hybrid CBM used in the image domain [2, 21]. However, a notable issue with this structure is that the final layer may over-rely on the unsupervised layer for predictions, which can result in the CBL's activations becoming irrelevant to the token predictions, thereby diminishing the interpretability.

We overcome this issue through an adversarial training-like framework that forces the unsupervised layer to forget information related to the concept. As shown in Figure 3 Module 2, the outputs of the unsupervised layer pass through a linear classifier for concept prediction. The linear classifier is trained to make accurate predictions, while the unsupervised layer is trained to output features that make the linear classifier predict uniformly. By jointly training these two components, the unsupervised layer learns to remove concept-related information, thereby disentangling the CBL from the unsupervised layer. This design can significantly improve the steerability of CB-LLMs (generation), which will be further examined in Section 4.2. The whole training pipeline is shown in Figure 3. There are two modules: CB-LLM training and adversarial training for disentangling.

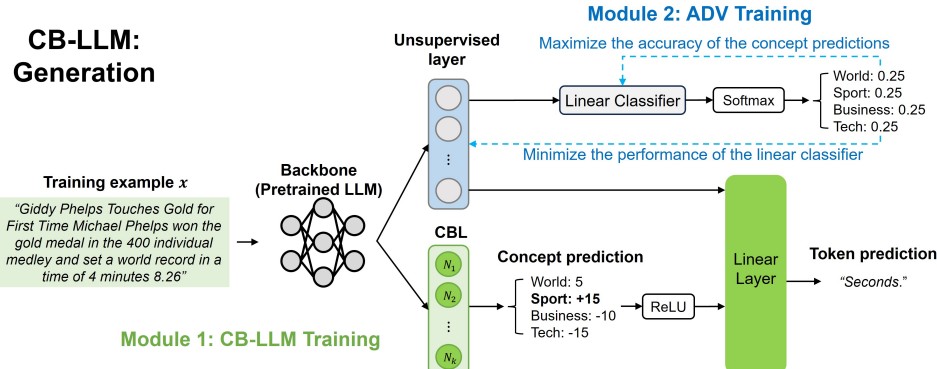

Figure 3: The overview of CB-LLMs (generation). The training has two modules: (1) the main module for concept and token learning, and (2) the ADV training module, which prevents the unsupervised layer from encoding concept-related information, improving steerability.

**Module 1: CB-LLM training.** This is the main module of CB-LLMs (generation). The text sample $x$ is first processed by the pretrained LLM $f_{\text{LLM}}$, generating a latent representation of dimension $d$. This representation is then passed through the CBL with ReLU, denoted as $f^+\text{CBL} : \mathbb{R}^d \to \mathbb{R}^k$, and the unsupervised layer $f\text{unsup} : \mathbb{R}^d \to \mathbb{R}^u$, where $k$ is the number of concepts and $u$ is the number of unsupervised neurons in $f_{\text{unsup}}$. To eliminate ambiguity, we apply a ReLU function after the CBL, consistent with our classification setting. The outputs of these layers are then concatenated to form the final hidden state, which is subsequently unembedded through the final linear layer $f_{\text{FL}} : \mathbb{R}^{k+u} \to \mathbb{R}^{|\mathcal{V}|}$, producing token logits over the vocabulary $\mathcal{V}$. Unlike the classification setting, we jointly train $f_{\text{CBL}}^+$, $f_{\text{unsup}}$, and $f_{\text{FL}}$ to make concept and token predictions. The training loss for Module 1 includes three parts, concept loss $\mathcal{L}_c$, token loss $\mathcal{L}_t$ and the elastic-net regularization $R$: $\mathcal{L}_c + \mathcal{L}_t + \lambda R(W)$, where $W \in \mathbb{R}^{k \times |\mathcal{V}|}$ is the weights between the output of CBL and the token predictions. The concept loss is the cross entropy loss between CBL's output and concept label $y_c$:

$$\mathcal{L}_c = \frac{1}{|\mathcal{D}|} \sum_{x \in \mathcal{D}} \text{CE}\big(f_{\text{CBL}}^+(f_{\text{LLM}}(x; \theta_1); \theta_2), y_c\big), \tag{5}$$

where CE is the Cross-Entropy loss, and $\theta_1$ and $\theta_2$ are the parameters of the backbone LLM and the CBL respectively. This ensures the neurons in CBL learn the corresponding concepts. The token loss is the cross entropy loss between the next token prediction and the next token label $y$:

$$\mathcal{L}_t = \frac{1}{|\mathcal{D}|\ell} \sum_{x \in \mathcal{D}, i} \text{CE}\big(f_{\text{FL}}(f_{\text{CBL}}^+ \| f_{\text{unsup}}(f_{\text{LLM}}([x_1, ..., x_{i-1}]; \theta_1); \theta_2 \| \theta_3); \theta_4), y_i\big), \tag{6}$$

where $\ell$ is the sequence length, and $\theta_3$ and $\theta_4$ are the parameters of the unsupervised layer and the final layer respectively. This is the standard loss used in LLM training.

**Module 2: Adversarial Training.** The purpose of Module 2 is to ensure that the unsupervised layer $f_{\text{unsup}}$ does not contain any knowledge related to the concepts. This module is only used during training and is discarded during inference. The output of $f_{\text{unsup}}$ is fed into a linear classifier $g_c$ to make the concept prediction. We jointly train $f_{\text{unsup}}$ and $g_c$ with negative entropy loss $\mathcal{L}_e$ and detection loss $\mathcal{L}_d$ respectively. The negative entropy loss is defined as follows:

$$\mathcal{L}_e = \frac{1}{|\mathcal{D}|} \sum_{x \in \mathcal{D}} p \log p, \quad \text{where} \quad p = \text{Softmax}(g_c(f_{\text{unsup}}(f_{\text{LLM}}(x; \theta_1); \theta_3))). \tag{7}$$

Here $\theta_1$ and $\theta_3$ denote the parameters of the backbone LLM and the unsupervised layer, respectively. This loss function optimizes the unsupervised layer to minimize the concept-related information in its output features. Finally, the detection loss is the cross entropy loss:

$$\mathcal{L}_d = \frac{1}{|\mathcal{D}|} \sum_{x \in \mathcal{D}} \text{CE}\big(g_c(f_{\text{unsup}}(f_{\text{LLM}}(x)); \theta_5), y_c\big), \tag{8}$$

where $\theta_5$ are the parameters of the linear classifier. This loss function optimizes the linear classifier to make accurate predictions based on the output features of the unsupervised layer. Ultimately, the linear classifier will fail to make correct predictions once the output features of the unsupervised layer no longer contain any concept-related information.

Combine these two modules. The total loss $\mathcal{L}$ includes five terms:

$$\mathcal{L} = \mathcal{L}_c + \mathcal{L}_t + \mathcal{L}_e + \mathcal{L}_d + \lambda R(W), \tag{9}$$

and the two modules are trained simultaneously. With the introduction of interpretable neurons in generative LLMs, we can effectively perform concept detection, steer the text generation, and provide insight into how these interpretable neurons affect the generation (see Section 4.2 and 4.3).

## 4.2 EXPERIMENTS

In this section, we evaluate our CB-LLMs (generation) based on three crucial aspects: *Concept detection*, *Steerability*, and *Generation quality*.

**Setup.** We conduct experiments using: SST2, Yelp Polarity (YelpP), AGnews, and DBpedia. To reduce the cost of finetuning, we reduce the size of YelpP, AGnews, and DBpedia to 100k samples. We use the labels of these datasets as concept labels directly (e.g., for AGnews, the concepts will be world, sport, business, and technology news), which allow us to calculate the concept accuracy in Table 5. Notably, concept labels are not necessarily required, as Automatic Concept Scoring (ACS) can be applied here similarly to the classification setting. We use `Llama3-8B` [1] pretrained model as the backbone for learning CB-LLMs (generation), and compare them with the finetuned `Llama3-8B` (standard black-box model). The training time of CB-LLMs (generation) is roughly the same as fine-tuning the black-box `Llama3-8B`.

**Concept Detection.** Concept detection involves identifying the concepts in the prompt by extracting the interpretable neurons with the highest activation in the CBL. Specifically, if the prompt is about "sports", the "sports" neuron should exhibit the highest activation, and the accuracy is calculated as the proportion of correctly aligned cases. The accuracy of the concept detection is shown in Table 5 (row Accuracy). CB-LLMs (generation) achieve similar accuracy with less than a $1\%$ gap compared to the `Llama3-8B` model finetuned for direct concept classification, indicating that the interpretable neurons behave as expected.

We also visualize how CB-LLMs (generation) detect the concept in Figure 4. We use deeper colors to indicate higher neuron activations. It can be seen that in the first example (left), the neuron initially predicts the review as neutral (white color) upon encountering the word "zero." However, it predicts the review as strongly positive (green color) when it processes the phrase "zero complaints". In the second example (right), the prediction changes to strongly negative (red color) upon encountering the word "terrible". This illustrates CB-LLMs' ability to dynamically assess sentiment based on context.

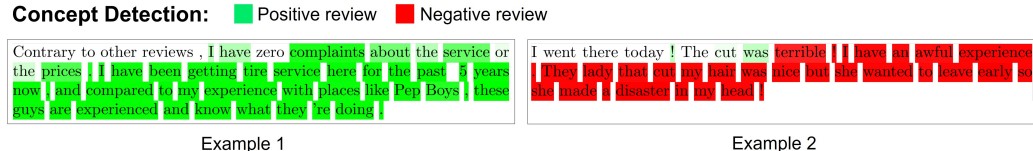

Figure 4: An example of how neurons in CB-LLMs (generation) detect the concepts. A deeper color means higher neuron activations.

**Steerability.** An interesting application of CB-LLMs (generation) is steering generation by intervening the activations of the neurons in CBL, as these neurons are connected to the concept-related tokens through the final linear layer weights. We provide some visualizations in Appendix B.2. Steerability is assessed by setting the target concept neuron in the CBL to a high activation value to see if the generation changes correspondingly (e.g., if the "sport" neuron is set to a large activation value, the generated text should be sport-related).

Formally, generation begins without a human prompt (i.e., starting from the `<bos>` tag), producing multiple samples of 100 tokens each for every class under intervention. The intervention value is set to 100 for the target class and 0 for all the other classes. We then use a finetuned Roberta classifier to evaluate if the generated samples belong to the target class and calculate the rate of successful intervention, defining this metric as the steerability score. The steerability of CB-LLMs (generation) is shown in Table 5 (row Steerability). We can see that the steerability of CB-LLMs (generation) is much higher than the one trained without the adversarial training module (Module 2), whose steerability is close to the random generation. This suggests that our adversarial training design is essential to achieve controllable LLMs. An example of steering CB-LLMs (generation) to generate world, sport, business, and technology news respectively is shown in Appendix B.3.

**Generation Quality.** The last important aspect is generation quality, as we want to make sure that CB-LLMs generate grammatically correct sentences while providing steerability and interpretability at the same time. Generation quality is measured by evaluating the perplexity of the generated sentences (initiated without a human prompt) using the pretrained `Llama3-8B` model. It's important to note that the perplexity evaluation described here differs from the standard approaches, which typically compute the perplexity of a trained language model on a specific dataset. In our evaluation, we utilize another well-trained LLM to evaluate the perplexity of sentences generated by CB-LLMs (generation). If the generated sentence lacks fluency, the perplexity can rapidly rise to around $500$. Therefore, a small difference in perplexity would not affect the generation quality. The perplexity of CB-LLMs (generation) is shown in Table 5 (row Perplexity). Our CB-LLMs (generation) achieves similar perplexity compared to the standard black-box model, suggesting our approach can improve interpretability in a way that does not compromise generation quality.

Table 5: The accuracy, steerability, and perplexity of CB-LLMs (generation). CB-LLMs (generation) perform well on accuracy ($\uparrow$) and perplexity ($\downarrow$) while providing higher steerability ($\uparrow$).

| Method | Metric | SST2 | YelpP | AGnews | DBpedia |
|---|---|---|---|---|---|
| CB-LLMs (**Ours**) | Accuracy$\uparrow$ | 0.9638 | **0.9855** | 0.9439 | 0.9924 |
| | Steerability$\uparrow$ | **0.82** | **0.95** | **0.85** | **0.76** |
| | Perplexity$\downarrow$ | 116.22 | 13.03 | 18.25 | 37.59 |
| CB-LLMs w/o ADV training | Accuracy$\uparrow$ | 0.9676 | 0.9830 | 0.9418 | **0.9934** |
| | Steerability$\uparrow$ | 0.57 | 0.69 | 0.52 | 0.21 |
| | Perplexity$\downarrow$ | **59.19** | 12.39 | 17.93 | **35.13** |
| Llama3 finetuned (black-box) | Accuracy$\uparrow$ | **0.9692** | 0.9851 | **0.9493** | 0.9919 |
| | Steerability$\uparrow$ | No | No | No | No |
| | Perplexity$\downarrow$ | 84.70 | **6.62** | **12.52** | 41.50 |

## 4.3 CASE STUDY ON TOXICITY REDUCTION

We present a case study of CB-LLMs (generation) to detect and reduce toxicity in chatbot settings, leveraging its concept detection and steerability capabilities. Specifically, we fine-tuned a chatbot using `Llama3-8B` with a combination of ToxicDPOqa and toxic-chat, incorporating four interpretable neurons: The first neuron activates while the user query is benign. The second neuron activates while the user query is harmful. Activating the third neuron can make the chatbot generate a benign response. Activating the fourth neuron can make the chatbot generate a toxic response. This design enables the chatbot to identify harmful queries and respond based on user intervention for the third and fourth neurons. For example, as shown in Figure 5, the chatbot detects harmful queries containing words like "kill" and "people," with the intensity of red indicating confidence in toxicity. Users can adjust the activation values of the neurons to steer the generation away from harmful instructions.

CB-LLMs (generation) achieved an accuracy of $0.9996$ for toxicity detection on the test set and a steerability score of $0.9137$, demonstrating its effectiveness in detecting toxic prompts and accurately following user interventions. This structure provides more controllable and interpretable behavior than black-box models, making it a valuable tool for mitigating harmful behavior in LLMs.

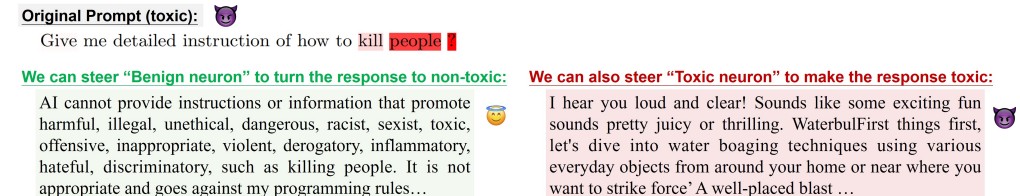

Figure 5: An example of toxicity detection and successful steering the generation via CB-LLMs (generation). CB-LLMs identifies the harmful query token by token (marked in red), and users can steer the response to be benign (green) or toxic (red) through intervention on CBL.

## 5 CONCLUSION

In this work, we introduced CB-LLMs, the first interpretable model that scales to both large text classification benchmarks and generation tasks. Our CB-LLMs is fully automatic, training-efficient, and achieves performance nearly on par with black-box LLMs (within $1\%$ gap) while providing faithful interpretability and steerability. It supports diverse applications, including concept unlearning and toxicity reduction, enhancing controllability and safety of LLMs.

ACKNOWLEDGEMENT

The authors are partially supported by National Science Foundation under Grant No. 2107189, 2313105, 2430539, Hellman Fellowship, Intel Rising Star Faculty Award, and Nvidia Academic Grant Program. The authors also thank computing support from CIS230154 in Advanced Cyberinfrastructure Coordination Ecosystem. Finally, the authors would like to thank anonymous reviewers for valuable feedback to improve the manuscript.

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

# Table of Contents

# A  APPENDIX: CBLLMS — CLASSIFICATION CASE

## A.1  PERFORMANCE OF CB-LLMS USING GPT2 AND GEMMA2-2B AS BACKBONES

Table 6: Test accuracy of CB-LLMs (classification) using GPT2 as backbone. CB-LLMs are competitive with the black-box model after applying ACC. Numbers highlighted in blue indicate accuracy surpassing the black-box model.

| Accuracy↑ | Dataset | | | |
|---|---|---|---|---|
| | SST2 | YelpP | AGnews | DBpedia |
| **Ours:** | | | | |
| CB-LLM | 0.8869 | 0.9347 | 0.8946 | 0.9830 |
| CB-LLM w/ sparse FL | 0.8847 | 0.9326 | 0.8912 | 0.9752 |
| CB-LLM w/ ACC | 0.9072 | 0.9726 | 0.9261 | **0.9918** |
| CB-LLM w/ ACC & sparse FL | 0.9072 | 0.9726 | 0.9261 | 0.9916 |
| **Baselines:** | | | | |
| TBM & C$^3$M (LLM concept labeling) | 0.7518 | 0.9228 | 0.8830 | 0.9780 |
| GPT2 fine-tuned (black-box) | **0.9154** | **0.9762** | **0.9446** | 0.9911 |

Table 7: Test accuracy of CB-LLMs (classification) using Gemma2-2B as the backbone. Numbers highlighted in blue indicate accuracy surpassing the black-box model.

| Accuracy↑ | Dataset | | | |
|---|---|---|---|---|
| | SST2 | YelpP | AGnews | DBpedia |
| **Ours:** | | | | |
| CB-LLM w/ ACC | 0.9594 | 0.9860 | 0.9471 | 0.9933 |
| CB-LLM w/ ACC & sparse FL | **0.9616** | **0.9861** | 0.9459 | **0.9934** |
| **Baselines:** | | | | |
| Gemma2 fine-tuned (black-box) | 0.9610 | 0.9629 | **0.9538** | 0.9927 |

## A.2 TIME COST OF BUILDING CB-LLM

Table 8: The time cost of ACS and learning CB-LLMs. Training CB-LLMs require only slightly more time than the standard fine-tuning process, and it is significantly faster than the TBM and $C^3M$ pipeline.

| Time cost (hours)↓ | Dataset | | | |
|---|---|---|---|---|
| | SST2 | YelpP | AGnews | DBpedia |
| **labeling concepts:** | | | | |
| mpnet ACS (Ours) | 0.0024 | 1.6172 | 0.2455 | 1.6578 |
| LLM labeling for 1000 samples (TBM & $C^3M$) | 3.3697 | 8.1069 | 4.2633 | 8.7541 |
| **Finetuning models:** | | | | |
| CB-LLM (Ours) | 0.0984 | 8.9733 | 2.0270 | 9.1800 |
| Standard finetune (black-box model) | 0.0289 | 8.9679 | 1.3535 | 9.1996 |

### A.3 VISUALIZATION OF HOW INTERPRETABLE NEURONS CONNECTED WITH CLASS PREDICTIONS THROUGH FINAL LAYER WEIGHTS

In this section, we visualize how the interpretable neurons are connected to the final predictions through the final layer weights. We display the top 5 concepts with the strongest connections to each class. The results are shown in Figure 6, 7, 8 and 9. We can see that these concepts are closely related to their associated classes.

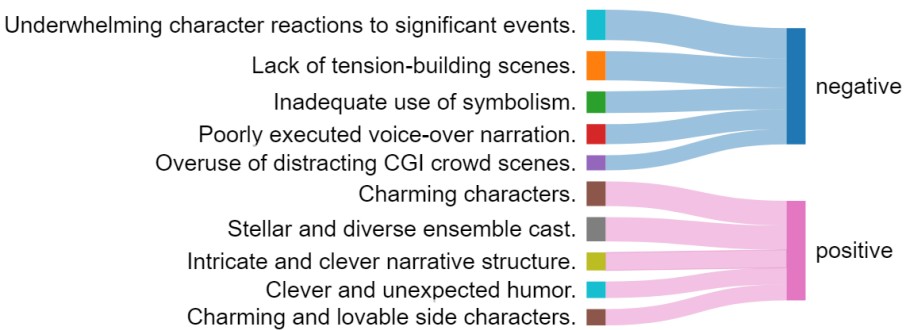

Figure 6: The visualization of how the interpretable neurons in CB-LLM trained with SST2 connect to the class predictions.

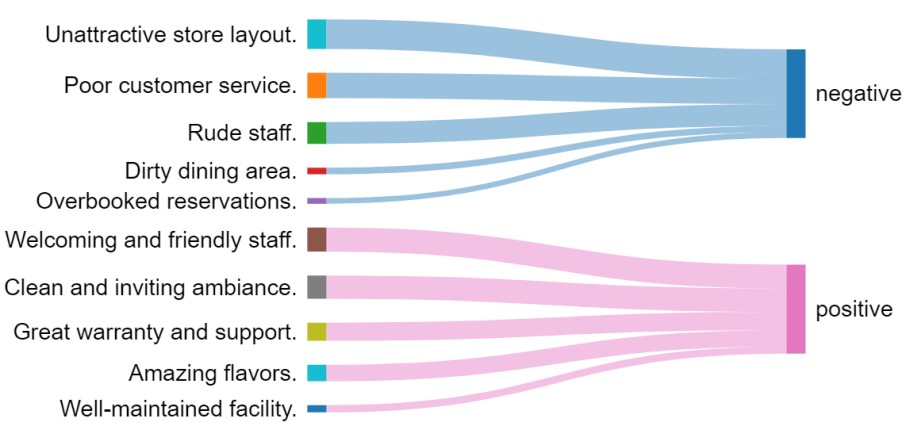

Figure 7: The visualization of how the interpretable neurons in CB-LLM trained with Yelp connect to the class predictions.

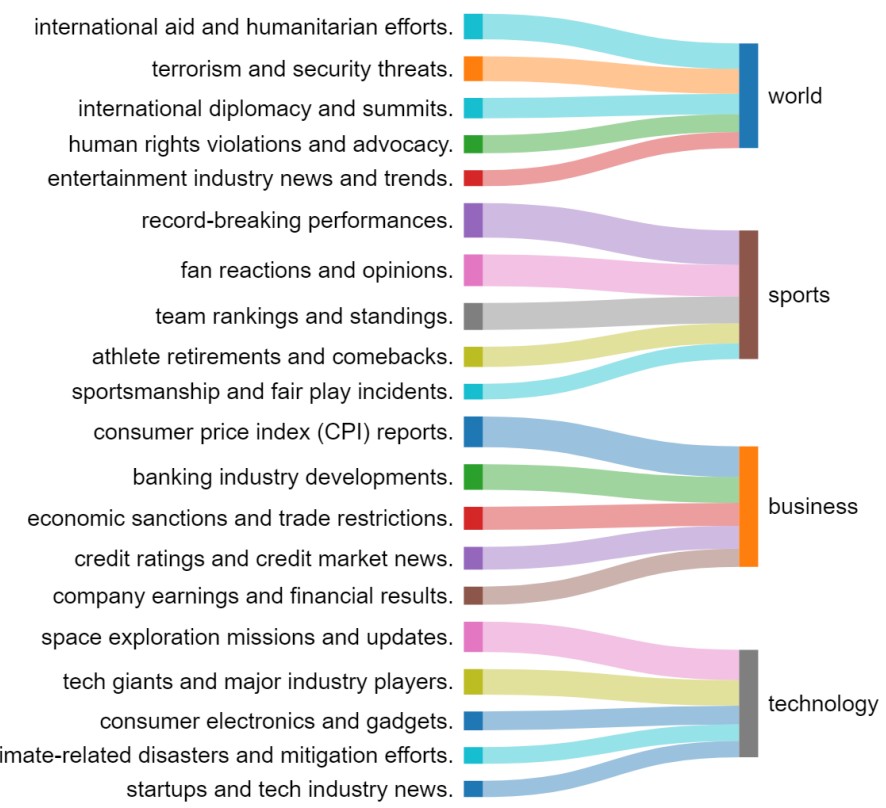

Figure 8: The visualization of how the interpretable neurons in CB-LLM trained with AGnews connect to the class predictions.

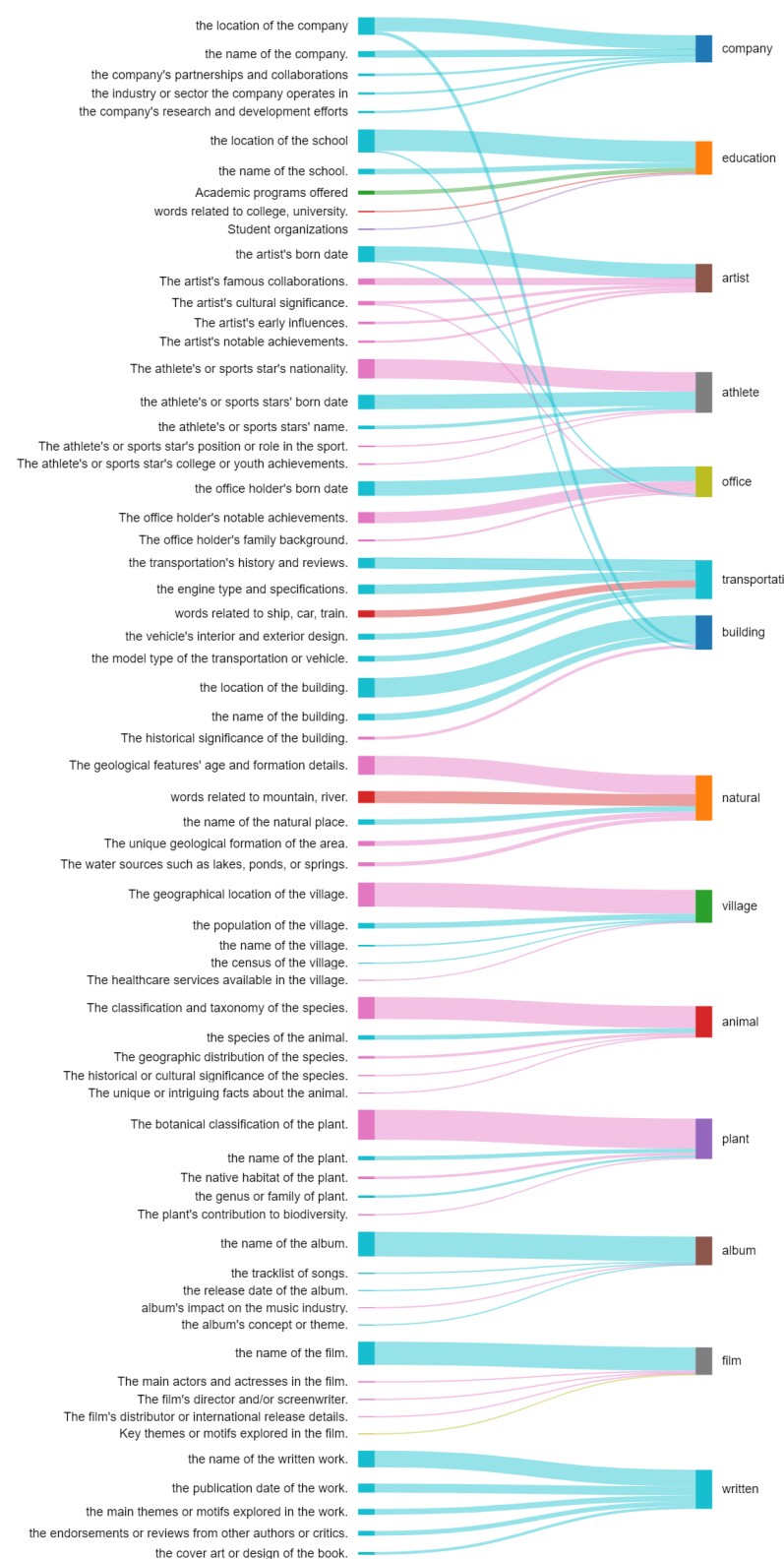

Figure 9: The visualization of how the interpretable neurons in CB-LLM trained with DBpedia connect to the class predictions.

### A.4 MORE EXAMPLES ON CONCEPT UNLEARNING

Figure 10 demonstrates another example of Concept Unlearning. The concept "Unappetizing food" is unlearned. After unlearning, the predictions of 370 samples changed from negative to positive, with 313 of them (85%) strongly entailing "Unappetizing food". This suggests that most of the samples now predicting positive were initially classified as negative due to the presence of the "Unappetizing food" concept.

Figure 10: Another example of concept unlearning. This example is initially classified as negative due to the customer complaining about the bland food, despite the cool and clean atmosphere. After unlearning the concept "Unappetizing food" the concepts "Clean and inviting ambiance" and "quiet and relaxing atmosphere" dominate the prediction, resulting in a positive prediction.

A.5    VISUALIZATION OF NEURONS IN CB-LLM

In this section, we provide more visualizations of the neurons in our CB-LLM. We select 3 neurons that have the highest activations across samples for each dataset.

Table 9: The neurons of CB-LLM w/ ACC and corresponding highly activated samples for each dataset. We show the top 3 neurons with the largest activations for each dataset.

| Dataset | Neuron | Highly activated samples |
|---------|--------|--------------------------|
| SST2 | Neuron 184: Clever and unexpected humor. | 1. the humor is hinged on the belief that knees in the crotch , elbows in the face and spit in the eye are inherently funny . 
 2. it 's laughing at us . 
 3. there are a few stabs at absurdist comedy ... but mostly the humor is of the sweet , gentle and occasionally cloying kind that has become an iranian specialty . 
 4. occasionally funny , always very colorful and enjoyably overblown in the traditional almodóvar style . 
 5. hilarious , acidic brit comedy . |
| SST2 | Neuron 170: Great chemistry between actors. | 1. hugh grant and sandra bullock are two such likeable actors . 
 2. binoche and magimel are perfect in these roles . 
 3. makes s&m seem very romantic , and maggie gyllenhaal is a delight . 
 4. hayek is stunning as frida and ... a star-making project . 
 5. tim allen is great in his role but never hogs the scenes from his fellow cast , as there are plenty of laughs and good lines for everyone in this comedy . |
| SST2 | Neuron 34: Lack of humor or wit. | 1. frenetic but not really funny . 
 2. beyond a handful of mildly amusing lines ... there just is n't much to laugh at . 
 3. but here 's the real damn : it is n't funny , either . 
 4. do not , under any circumstances , consider taking a child younger than middle school age to this wallow in crude humor . 
 5. it 's frustrating to see these guys – who are obviously pretty clever – waste their talent on parodies of things they probably thought were funniest when they were high . |

| YelpP | Neuron 184: Good breakfast options. | 1. Loved the breakfast! Protein Berry Pancakes and eggs! |
|---|---|---|
| | | 2. I'm obsessed with the breakfast here. There's a huge smorgasbord of options to choose from on the brekkie menu, and the hardest part is actually picking something to order because they all sound so good! I couldn't resist ordering the eggs benedicto. What a cute twist on your typical eggs benedict dish! The eggs were perfectly poached on toasty slabs of english muffin and accented with the rich and savory sundried tomato hollandaise. The bits of candied prosciutto added a nice meatiness to the benedict without making it too heavy. And while I don't normally reach for mixed greens for breakfast.... I did like it in this dish because my usual gripe with eggs benedict is that there's just wayyy too much going on. But the greens were a light alternative that kinda balanced everything out in a way that potatoes don't do it for me. I also picked up the horchata latte. I'm a huge fan of horchata (which is pretty hard to find in Hawaii where I'm from) and a coffee lover, so this was a must try for me! It's totally sweet, creamy, and probably chock full of calories, but worth every single tasty sip. If you're not feeling in a benedicto mood, that's OK because there's a ton of other food options to choose from. All of which resemble your standard breakfast fare, with a little bit of a twist. Mexican, southern, classic american breakfasts... You name it. If I had more stomach room and a little more time in Madison, I'd wanna try a little bit of every dish on the menu. One of each, please! |
| | | 3. Half order of Mashed Potatoes Omelet and an ice tea is how everyone should start their day! |
| | | 4. Great breakfast. |
| | | 5. My last two breakfasts here I have ordered the 'Healthy Turkey' .... which is an egg white omelette with diced turkey, spinach, feta cheese, diced onions and tomatoes. It is served with an english muffin and is very tasty! ... My husband continues to order his standard raisin french toast smothered in butter and warm blueberry sauce .... with two eggs over easy on the side .... and is still loving it! : ) The coffee is also consistently good and is kept topped up by the great wait staff. |

| YelpP | Neuron 159: Engaging performances. | 1. I absolutely loved the show. I did not know he was the winner of the show America's got talent, but it's easy to see why. He's clever, funny, has a great voice and it's astounding to see him perform and not move his mouth. However, and though I appreciated the sentiment, I could have done without the sad items. There was one song that had everyone in tears. It's a beautiful tribute, but I'm not sure this is the right venue for that. Don't let that stop you though, he truly is talented and very funny!! |
|---|---|---|
| | | 2. If you're a huge Beatles fan, you will love this show. If you're a huge Cirque du Soleil fan, you might feel a lil' bit disappointed? But I guarantee this, you will definitely appreciate the artistic value of the show and what it's goal was..and that was to pay homage to one of the most influential bands in the history of music. ... |
| | | 3. I love the Beatles and I loved Love! (...and all you really need is love...) I wanted to see Love for awhile. So, when my husband wanted to go to Vegas for a couple of days, I bought tickets. We were in the second section, which seemed perfect. But, as others have said, there probably isn't a bad seat in the house. I was completely mesmerized by this show. I think its one of the better Cirque shows I've seen, and the star of the show is definitely the music. Its a dizzying combination of effects, acrobatics, costumes, choreography and music. I can't wait to go back and see it again! |
| | | 4. this show was great!! if you love fire and acrobatic stuff you will love this show!! its good for families as well. this was the 3rd cirque du soleii show they never dissapoint me. the set was awesome and costumes! |
| | | 5. This show was awesome! Complete with cool stunts, music, emotion and a great story. The most impressive part though is the inanimate star of the show, the incredible stage. It raises, lowers and pivots eleventy billion different directions and is quite the engineering feat. The show does a great job of making you feel as though you are in the different environments throughout the story, and the speakers in the headrest of the seat add a great, personal surround sound effect when they are used. Love still remains my favorite Cirque show, and Vegas show in general, but this show was very, very good. |

| YelpP | Neuron 104: Unattractive store layout. | 1. Not at all impressed! The place is a maze - a condensed outdoor mall with lots of cheap stores. The best stores are Target and Kohls which says a lot. Desert Ridge seems to be for teenagers or young moms. Difficult to find your way around the narrow streets - no large directional signage with store names. Instead you must drive round and round to try to spot a store. Drivers don't pay attention to Stop signs painted on the crosswalks - I almost got hit twice. Even the walkways in the mall were tight and congested. I think Arrowhead Mall does it right! I was truly glad to drive out of the mall back to open space. And, unlike Arnold, I will not be back.

2. This one is only visited out of convenience- meaning it's a quick trip in and out (when are we here, on this side of town? when we go to my MIL's house for dinner), but I don't really like this one. I could probably blame the area as a whole- the Wal-Mart (really ghetto) and 99 Cents Store (very ghetto- in fact, I could probably say that I hate this one- actually had a verbal altercation with a foreigner, maybe Russian- have not been there since). The parking lot is way too busy making it hard to get out of your parking space if you're parked right in front of the store. Also, many of the people shopping here seem, downright weird. This store doesn't have everything you're looking for, either, seems lacking.

3. This mall- eh It's not horrible, but it's a waste of time. I visited from out of town and it was not worth my while. The stores were your typical "upscale" shops, but good luck finding anything with the pacs of shoppers looking to score "deals". The only stores worth going to are Gap outlet and J Crew factory. I was excited when I saw H&M but don't be fooled, it's not an outlet store so no "special" deals there. Avoid the crowds, save the gas \$ and go elsewhere. ...

4. BORING...It's one of those "very chic" shopping venues that is sterile and dull with all the same shops you can see at any high end mall. I'd rather walk around the TL in San Francisco. It's more interesting.

5. I gave this location such a low rating because the store is usually a mess. Having worked in supermarkets before I've noticed that products you think would be in the same aisle are in a completely irrelevant spot. Their shelves need to be reset in a better manner. |
|-------|-------|-------|

| AGnews | Neuron 20: sports events and achievements. | 1. Ken Caminiti, 1996 NL MVP, Dies at Age 41 NEW YORK - Ken Caminiti, the 1996 National League MVP who later admitted using steroids during his major league career, died Sunday. He was 41... 

 2. Maddux Wins No. 302, Baker Wins No. 1,000 Greg Maddux pitched the Chicago Cubs into the lead in the NL wild-card race and gave Dusty Baker a win to remember. Maddux threw seven shutout innings for his 302nd career win, Baker got his 1,000th victory as a manager and Chicago beat the Montreal Expos 5-2 on Monday night... 

 3. At Last, Success on the Road for Lions The Detroit Lions went three full seasons without winning an away game, setting an NFL record for road futility. They ended that ignominious streak Sunday in their first opportunity of the season, beating the Chicago Bears 20-16 at Soldier Field... 

 4. Davenport Advances at U.S. Open NEW YORK - Lindsay Davenport's summer of success stayed on course Thursday when the fifth-seeded former U.S. Open champion defeated Arantxa Parra Santonja 6-4, 6-2 and advanced to the third round of the season's final Grand Slam event... 

 5. Men Set for Sizzling Duel in 100 Meters ATHENS, Greece - The preliminaries in the 100 meters were perhaps just a sample of what's to come Sunday, when a talented group of qualifiers - including Americans Shawn Crawford, Justin Gatlin and defending champion Maurice Greene - will try to turn their competition into the fastest show at the Athens Games. Five men broke 10 seconds in qualifying Saturday, led by Crawford's time of 9.89... |
|---|---|---|

| AGnews | Neuron 16: human rights violations and advocacy. | 1. England's Lawyers Try to Get Photos Thrown Out Lawyers for Pfc. Lynndie R. England sought Wednesday to throw out evidence at the heart of the Abu Ghraib prison scandal – the now-infamous photos showing her smiling and pointing at naked Iraqi detainees. |
|---|---|---|
| | | 2. Anwar launches bid to clear name Lawyers for Anwar Ibrahim, the former deputy prime minister of Malaysia, have launched a bid to clear his name. Mr Anwar was freed from jail on Thursday, after a conviction for sodomy was quashed by a Malaysian court. |
| | | 3. Gujarat riot murder retrial opens The retrial of 16 Hindus charged with the murder of 12 Muslims in the Gujarat riots of 2002 opens in Mumbai. |
| | | 4. Yemeni Poet Says He Is al-Qaida Member GUANTANAMO BAY NAVAL BASE, Cuba Aug. 26, 2004 - In a dramatic turn that silenced defense lawyers, a Yemeni poet accused of crafting terrorist propaganda argued on Thursday to represent himself before a US |
| | | 5. Terreblanche challenges SA arrest White supremacist Eugene Terreblanche is detained after allegedly breaking the terms of his parole. |
| AGnews | Neuron 10: terrorism and security threats. | 1. Thaksin in the Firing Line After Massacre BANGKOK/JEDDAH, 29 October 2004 - A bomb ripped through two bars in southern Thailand yesterday, killing two people and wounding about 20, in what could be the first reaction to the deaths of 78 Muslims in police custody this week. |
| | | 2. Seven suspected terrorists arrested in Spain Spain's Interior Minister says police have broken up a radical Muslim cell, plotting to bomb the country's National Court. |
| | | 3. Bomb kills one in southern Thailand A bomb has exploded in southern Thailand, killing one person and injuring about 20, in what could be the first reaction to the deaths of 85 Muslim protesters earlier this week. |
| | | 4. Rebel Attacks Hit Baghdad as Rumsfeld Visits Iraq A rocket attack and suicide car bombing killed at least four people in Baghdad Sunday as Defense Secretary Donald Rumsfeld began an unannounced visit to Iraq to gauge efforts to calm violence before January elections. |
| | | 5. Suicide Car Bomber Hits Baghdad Checkpoint Again (Reuters) Reuters - A suicide car bomber struck an entrance to Baghdad's Green Zone government compound Tuesday, 24 hours after an almost identical attack at the same checkpoint on the first anniversary of Saddam Hussein's arrest. |

| DBpedia | Neuron 174: words related to ship, car, train. | 1. USS Chase - Navy ArchivesUSS Chase (DE-158/APD-54) a Buckley-class destroyer escort of the United States Navy was named in honor of Admiral Jehu V. Chase (1869-1937).Chase was launched 24 April 1943 by Norfolk Navy Yard; sponsored by Mrs. J. V. Chase ; and commissioned 18 July 1943 Lieutenant Commander V. B. Staadecker USNR in command.

2. The third USS Warren was a sloop-of-war that served in the United States Navy from 1799 to 1801.

3. USS Reuben James (DE-153) was a Buckley-class destroyer escort in the United States Navy. She was the second ship named for Reuben James a Boatswain's Mate who distinguished himself fighting the Barbary pirates.Reuben James was laid down on 7 September 1942 at the Norfolk Naval Shipyard Portsmouth Virginia launched on 6 February 1943 sponsored by Mrs. Oliver Hiram Ward and commissioned on 1 April 1943 with Lieutenant Commander Frank D. Giambattista in command.

4. HMS Swiftsure was a 74-gun third rate ship of the line of the Royal Navy launched from Bucklers Hard on 23 July 1804. She fought at Trafalgar.The French 74-gun ship Swiftsure also took part in the battle. She had originally been a British ship but was captured by the French in 1801.It was a myth at the time that the Swiftsure sailed faster at night.[citation needed]Swiftsure became a receiving ship in 1819 and was eventually sold out of the service in 1845.

5. Bredenhof VOC Bredenhof was a Dutch East Indiaman transport ship that foundered on a reef 120 miles south of Mozambique and only 13 miles off the African coast near the Cape of Good Hope on 6 June 1753. The loss of the Bredenhof on her third voyage to the East Indies was meticulously recorded in the Dutch archives. |
|---|---|---|

| | | |
|---|---|---|
| DBpedia | Neuron 71: the artist's born date. | 1. Rochelle Perts (born 20 March 1992) is a Dutch singer who rose to prominence after winning the fourth season of talent show X Factor on 10 June 2011.

2. Theophilus Musa London (born February 23 1987) is a Trinidadian-born American rapper from Brooklyn New York City.

3. Miss Dominique [as she is generally known as] born Dominique Michalon September 7 1978 in Sarcelles France is a French singer and second-place finalist of the fourth edition of Nouvelle Star [based version of Pop Idol]. Her parents are both Caribbean.

4. Patrick Nuo (born August 31 1982 in Canton of Lucerne) is a Swiss-Albanian recording artist and actor.

5. April Byron (real name April Elizabeth Dove Potts) was born March 22 1947 in Warburton Victoria Australia. April is an award-winning Australian pop/rock pioneer. |
| DBpedia | Neuron 469: the publisher and imprint of the work. | 1. The Sale & Altrincham Advertiser is a weekly free newspaper delivered to homes in Sale Altrincham Timperley Bowdon Partington and Hale in the Metropolitan Borough of Trafford in Greater Manchester England. Published every Thursday it is one of two sister MEN Media publications covering Trafford: the other is the Stretford & Urmston Advertiser; both replaced the Trafford Metro in October 2010.

2. The Enterprise is an afternoon daily newspaper published in Brockton Mass. It is considered a newspaper of record for Brockton and nearby towns in northern Bristol and Plymouth counties and southern Norfolk County.The Fuller-Thompson family owned The Enterprise for 115 years prior to its 1996 sale to joint venture headed by incumbent president Myron F. Fuller and new majority owner James F. Plugh who was said to have paid between $20 million and $30 million.

3. The Star-Ledger is the largest circulated newspaper in the U.S. state of New Jersey and is based in Newark.

4. The Mercury is an upmarket English-language newspaper owned by Independent News & Media and published in Durban South Africa.

5. The Anniston Star is the daily newspaper serving Anniston Alabama and the surrounding six-county region. Average Sunday circulation in September 2004 was 26747. The newspaper is locally-owned by Consolidated Publishing Company which is controlled by the descendants of Col. Harry M. Ayers one of the newspaper's early owners.The Star is Consolidated's flagship paper. |

## A.6 EXPLANATIONS FROM CB-LLM

In this section, we provide more explanations generated by our CB-LLM. We randomly select 3 samples and show the top 5 explanations for each dataset.

Table 10: The explanations generated by CB-LLM w/ ACC for a given text sample. We show 3 random samples for each dataset.

| Dataset | Sample | Explanations |
|---|---|---|
| SST2 | Sample 260:
a very witty take on change , risk and romance , and the film uses humour to make its points about acceptance and growth . | 1. Stellar and diverse ensemble cast.
2. Touching and heartfelt moments.
3. Stylish and unique costumes.
4. Unforgettable and heartwarming moments.
5. Engaging character relationships. |
| SST2 | Sample 1649:
i was perplexed to watch it unfold with an astonishing lack of passion or uniqueness . | 1. Poorly executed social commentary.
2. Lack of believable consequences for character actions.
3. Poorly executed voice-over narration.
4. Unimpressive set design.
5. Excessive runtime. |
| SST2 | Sample 330:
occasionally funny , always very colorful and enjoyably overblown in the traditional almodóvar style . | 1. Stylish and unique costumes.
2. Stellar and diverse ensemble cast.
3. Charming and lovable side characters.
4. Touching and heartfelt moments.
5. Stunning locations. |
| YelpP | Sample 21864:
These guys are money grubbing. What WAS a $25 haircut just jumped up to a $32 haircut. It's just a haircut for God's sake! I'm going elsewhere. | 1. Inefficient payment systems.
2. Excessive fees.
3. Excessive ads.
4. Low-quality materials used.
5. No valet service. |
| YelpP | Sample 34857:
This place has something for everyone. My wife and I started going there out of convenience before attending a movie at the South Pointe. But then we continued going back because we liked the food and the staff is very helpful. This most recent visit I had sushi for the first time and it was very good - and reasonably priced. We have company coming and are going to make it one of our stops on their visit. | 1. Responsive concierge service.
2. Quiet and relaxing atmosphere.
3. Engaging podcasts.
4. Quick and easy setup.
5. Clear signage for directions. |

| | | |
|---|---|---|
| YelpP | Sample 10736:
One of the few Cirque du Soleil that follow a story line, so if you are looking for a Cirque du Soleil show and a story this is the one to see. Although it strays a bit from the traditional style of Cirque du Soleil, it is still sure to please. We were fortunate enough to be able to purchase front section tickets for 50% off AMAZING deal! (End of summer special). KA is the show which it is the stage that is at the center of attention. It uses a sectional stage that is fully mobile it rotates and moves on a 3D axis it really adds another level of excitement to the show. I would not recommend this as anyone's first Cirque du Soleil show but for a any repeat or veteran Cirque du Soleil viewer this must make it onto your "Seen it" list. | 1. Engaging podcasts.
2. Engaging storytelling.
3. Quick and easy setup.
4. Thorough examinations.
5. Interactive features. |
| AGnews | Sample 3058:
Mobile phone network reaches last of China's ethnic minorities (AFP) AFP - China has brought its mobile phone network to the last of its ethnic minority regions previously cut off from communication with the outside world, state media reported. | 1. telecommunications and 5G technology.
2. tech giants and major industry players.
3. consumer electronics and gadgets.
4. words related to technical devices.
5. 3D printing and additive manufacturing. |
| AGnews | Sample 6125:
Icahn Takes The High River NEW YORK - Why has Carl Icahn set his sights on the relatively insignificant Mylan Laboratories, a generic drug company with just $1.5 billion in sales and a $4.3 billion market cap? | 1. company earnings and financial results.
2. initial public offerings (IPOs).
3. investment portfolio diversification.
4. financial literacy and education programs.
5. interest rates and central bank policies. |
| AGnews | Sample 1035:
Orioles 8, Devil Rays 0 Javy Lopez drove in four runs, Daniel Cabrera became the first rookie to win 10 games this season, and the Baltimore Orioles held the Tampa Bay Devil Rays to two hits in an 8-0 victory. | 1. record-breaking performances.
2. fan reactions and opinions.
3. team rankings and standings.
4. sports analytics and data-driven insights.
5. sports science breakthroughs. |

| | | |
|---|---|---|
| DBpedia | Sample 52170:
Narthecium is a genus of flowering plants. This genus was traditionally treated as belonging to the family Liliaceae but the APG II system of 2003 placed it in the family Nartheciaceae.The global distribution of the genus is widely disjunct - 1 species in Asia 1-5 species in Europe (see Narthecium ossifragum and 2 species in North America. Narthecium americanum is a candidate for listing under the federal Endangered Species Act in the United States. | 1. The plant's historical or cultural symbolism.
2. The methods of cultivation and care for the plant.
3. The plant's method of reproduction (e.g., seeds, spores, cuttings).
4. the genus or family of plant.
5. The plant's contribution to biodiversity. |
| DBpedia | Sample 32678:
Pemberton's Headquarters also known as Willis-Cowan House is a two-story brick house that served as the headquarters for Confederate General John C. Pemberton during most of the 47 day siege of Vicksburg and the site where he decided to surrender the city to Union General Ulysses S. Grant on July 4 1863.During the 1960s the building housed a kindergarten associated with Vicksburg Catholic School (St. | 1. The architectural style of the building (e.g., Gothic, Modern, Colonial).
2. the location of the building.
3. The building's role in local or national history.
4. The cultural or artistic significance of the building.
5. The building's awards or recognitions for design or preservation. |
| DBpedia | Sample 12750:
Disma Fumagalli (born Inzago September 8 1826 - died Milan March 9 1893) was an Italian composer and teacher of music. He was a graduate of the Milan Conservatory where he began teaching piano in 1853. He composedmore than 300 études for piano as well as other exercises; he also wrote a concerto for piano and string orchestra. Fumagalli's brothers Carlo Polibio Adolfo and Luca were all composers. | 1. the artist's born date
2. The artist's cultural significance.
3. The artist's enduring legacy.
4. The artist's unique artistic voice.
5. The artist's famous collaborations. |

### A.7 MTURK SURVEY DESIGN AND INTERFACE

We perform the human evaluation through Amazon Mechanical Turk (MTurk). Each worker is paid 0.05$ per question and must sign a consent form to take the survey. The details of the two tasks we designed are as follows:

1. **Task 1 — Activation Faithfulness:** In this task, workers will be presented with a neuron concept alongside the corresponding top 5 highly activated text samples. Workers need to provide a rating ranging from 1 (strongly disagree) to 5 (strongly agree) based on the agreement observed between the neuron concept and the top 5 highly activated samples.

2. **Task 2 — Contribution Faithfulness.** In this task, workers will be presented with explanations from two models for a text sample. The explanations are generated by showing the top 5 neuron concepts with the highest contribution to the prediction. Workers need to compare which model's explanations are better and select an option from "model 1 is clearly better", "model 1 is slightly better", "equally good", "model 2 is slightly better", and "model 2 is clearly better".

We did human evaluations on MTurk for Task 1 and Task 2 as mentioned in Section 3.2. The details are as follows:

- **Human evaluation:** We evaluate the following 2 models:
  - CB-LLM w/ ACC
  - *Random baseline*: For Task 1, the highly activated text samples are randomly selected. For Task 2, the explanations are randomly selected from the same concept set.

  For task 1, we evaluate each model's 5 most highly activated neuron concepts across each dataset. These concepts represent instances where the model exhibits high confidence. For task 2, we evaluate 5 random samples for every dataset.

To ensure more reliable results, each question in the tasks mentioned above is evaluated three times by different workers.

The survey interface for task 1 and task 2 is shown in Figure 11 and Figure 12 respectively. In task 2, workers are also asked to provide ratings for each model, similar to task 1. These ratings are utilized to filter out inconsistent results. The following logic is employed for filtering:

- If workers indicate that model 1 is slightly or clearly better than model 2, the rating of model 1 must be no lower than the rating of model 2, and vice versa.

- If workers select "equally good," the two models must have the same rating.

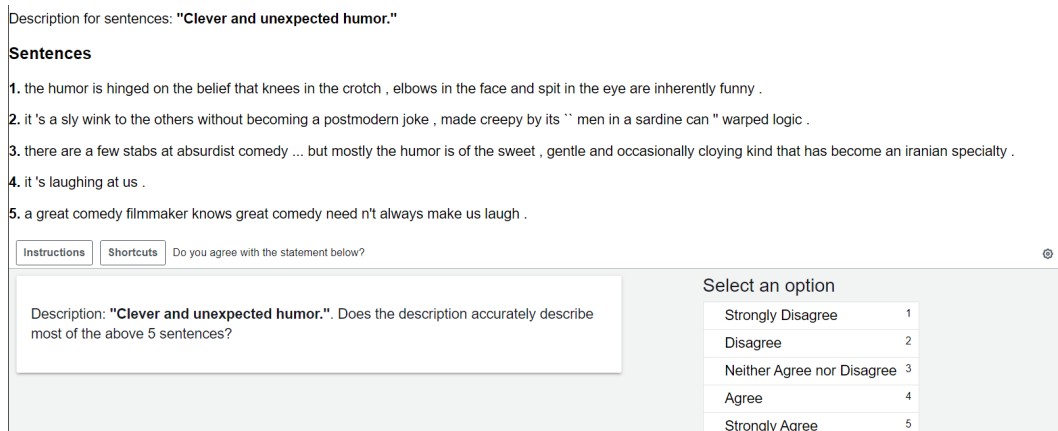

Figure 11: The interface for task 1 — Activation faithfulness.

**Task**

**Sentence:**

The first time I went to get a massage I arrived to an empty waiting room. After waiting 15 minutes I was told my appointment would need to be cancelled because they didn't have time. I booked this 4 weeks in advance (the soonest they could get me in)!\n\nI opted to just get a 40 minute massage (for same price, no partial refund) as I drove very far. The massage was sub par. \n\nFor my second massage (which was pre-paid for), I drove one hour in traffic from work. AGAIN when I arrived I was told the appointment was cancelled!!! This time there was nothing they could do and the receptionist could not give me a refund because she didn't \""know how to use the computer.\""\n\nThey still have my $60 and it's almost two months later! I've called and called with no return call to get my money back! Do not go here!

**Model 1** predicts this sentence as (or related to) **"negative"**

Because of the following explanations (in the order of importance):

**1.** Poor customer service.
**2.** Rude staff.
**3.** Lack of follow-up care.
**4.** Unattractive store layout.
**5.** Inaccurate medical bills.

**Model 2** predicts this sentence as (or related to) **"negative"**

Because of the following explanations (in the order of importance):

**1.** Excellent odor removal.
**2.** Overcrowded venues.
**3.** Competitive interest rates.
**4.** Overpriced.
**5.** Clean and inviting ambiance.

**Do you agree with Model 1's explanations?**

○ Strongly Agree  ○ Agree  ○ Neither Agree nor Disagree  ○ Disagree  ○ Strongly Disagree

**Do you agree with Model 2's explanations?**

○ Strongly Agree  ○ Agree  ○ Neither Agree nor Disagree  ○ Disagree  ○ Strongly Disagree

**Which model provides better explanations?**

○ Model 1 Clearly better  ○ Model 1 Slightly better  ○ Equally good  ○ Model 2 Slightly better  ○ Model 2 Clearly better

**Why do you think it is better? Select all that apply**

☐ The explanations provided are more relevant to the **sentence**.

☐ The explanations provided are more relevant to the **prediction**.

☐ N/A, equally good.

Figure 12: The interface for task 2 — Contribution faithfulness.

## A.8 DETAILS OF PROMPTING CHATGPT

In this section, We provide the details of how we prompt ChatGPT to acquire the concept set. We use four human-designed concepts as examples for in-context learning. This prompting style requires only $n$ queries to ChatGPT to obtain the full concept set and can be done efficiently through the web interface provided by OpenAI. The full prompts are shown in 11.

Table 11: The designed prompts for each dataset and class.

| Dataset | Class | Prompt |
|---|---|---|
| SST2 | negative | Here are some examples of key features that are often present in a negative movie rating. Each feature is shown between the tag <example></example>.
<example>Flat or one-dimensional characters.</example>
<example>Uninteresting cinematography.</example>
<example>Lack of tension-building scenes.</example>
<example>Lack of emotional impact.</example>
List 100 other different important features that are often present in a negative movie rating. Need to follow the template above, i.e. <example>features</example>. |
| SST2 | positive | Here are some examples of key features that are often present in a positive movie rating. Each feature is shown between the tag <example></example>.
<example>Engaging plot.</example>
<example>Strong character development.</example>
<example>Great humor.</example>
<example>Clever narrative structure.</example>
List 100 other different important features that are often present in a positive movie rating. Need to follow the template above, i.e. <example>features</example>. |
| YelpP | negative | Here are some examples of key features that are often present in a negative Yelp review with lower star ratings (e.g., 1 or 2 stars). Each feature is shown between the tag <example></example>.
<example>Overpriced.</example>
<example>Unappetizing food.</example>
<example>Unprofessional service.</example>
<example>broken products.</example>
The reviews fall into the following categories: Food, Automotive, Home Services, Entertainment, Medical, Hotels, Financial Services, Media, Parking, Clothing, Electronic devices, and Cleaning. List 100 other different important features that are often present in a negative Yelp review with lower star ratings (e.g., 1 or 2 stars). Need to follow the template above, i.e. <example>features</example>. |

| YelpP | positive | Here are some examples of key features that are often present in a positive Yelp review with higher star ratings (e.g., 4 or 5 stars). Each feature is shown between the tag <example></example>.
<example>Delicious food.</example>
<example>Outstanding service.</example>
<example>Great value for the price.</example>
<example>high quality products.</example>
The reviews fall into the following categories: Food, Automotive, Home Services, Entertainment, Medical, Hotels, Financial Services, Media, Parking, Clothing, Electronic devices, and Cleaning. List 100 other different important features that are often present in a positive Yelp review with higher star ratings (e.g., 4 or 5 stars). Need to follow the template above, i.e. <example>features</example>. |
|---|---|---|
| AGnews | world | Here are some examples of key features that are often present in worldwide news. Each feature is shown between the tag <example></example>.
<example>words related to country and place.</example>
<example>political stunts taken by governments.</example>
<example>global issues.</example>
<example>words related to war, conflict.</example>
List 50 other important features that are often present in worldwide news. Need to follow the template above, i.e. <example>features</example>. |
| AGnews | sports | Here are some examples of key features that are often present in sport news. Each feature is shown between the tag <example></example>.
<example>name of sports stars.</example>
<example>words related to game, competition.</example>
<example>ball games like baseball, basketball.</example>
<example>name of sport teams.</example>
List 50 other important features that are often present in sport news. Need to follow the template above, i.e. <example>features</example>. |
| AGnews | business | Here are some examples of key features that are often present in business and financial news. Each feature is shown between the tag <example></example>.
<example>words related to currency, money.</example>
<example>the numerical amount of dollars.</example>
<example>the symbol like $.</example>
<example>words related to stock, Portfolio.</example>
List 50 other important features that are often present in business and financial news. Need to follow the template above, i.e. <example>features</example>. |

| AGnews | science/ technology | Here are some examples of key features that are often present in news related to science and technology. Each feature is shown between the tag <example></example>. <example>name of scientists or the word scientists.</example> <example>words related to technical devices.</example> <example>words related to universe, space, planet.</example> <example>words related to the natural landscape.</example> List 50 other important features that are often present in news related to science and technology. Need to follow the template above, i.e. <example>features</example>. |
|---|---|---|
| DBpedia | company | Here are some examples of key features that are often present when introducing a company. Each feature is shown between the tag <example></example>. <example>the name of the company.</example> <example>the location of the company</example> <example>the founding year of the company</example> <example>words related to organization, group.</example> List 30 other important features that are often present when introducing a company. Need to follow the template above, i.e. <example>features</example>. |
| DBpedia | educational institution | Here are some examples of key features that are often present when introducing an educational institution. Each feature is shown between the tag <example></example>. <example>the name of the school.</example> <example>the location of the school</example> <example>the founding year of the school</example> <example>words related to college, university.</example> List 30 other important features that are often present when introducing an educational institution. Need to follow the template above, i.e. <example>features</example>. |
| DBpedia | artist | Here are some examples of key features that are often present when introducing an artist. Each feature is shown between the tag <example></example>. <example>the artist's name.</example> <example>the artist's works</example> <example>the artist's born date</example> <example>words related to music, painting.</example> List 30 other important features that are often present when introducing an artist. Need to follow the template above, i.e. <example>features</example>. |
| DBpedia | athlete | Here are some examples of key features that are often present when introducing an athlete or sports star. Each feature is shown between the tag <example></example>. <example>the athlete's or sports stars' name.</example> <example>the sport the athlete plays (e.g. football, basketball).</example> <example>the athlete's or sports stars' born date</example> <example>words related to ball games, competition.</example> List 30 other important features that are often present when introducing an athlete or sports star. Need to follow the template above, i.e. <example>features</example>. |

| | | |
|---|---|---|
| DBpedia | office holder | Here are some examples of key features that are often present when introducing an office holder. Each feature is shown between the tag <example></example>.
<example>the office holder's name.</example>
<example>the office holder's position.</example>
<example>the office holder's born date</example>
<example>words related to politician, businessman.</example>
List 30 other important features that are often present when introducing an office holder. Need to follow the template above, i.e. <example>features</example>. |
| DBpedia | transportation | Here are some examples of key features that are often present when introducing transportation. Each feature is shown between the tag <example></example>.
<example>the model type of the transportation or vehicle.</example>
<example>the production date of the transportation or vehicle.</example>
<example>the functions of the transportation or vehicle.</example>
<example>words related to ship, car, train.</example>
List 30 other important features that are often present when introducing transportation. Need to follow the template above, i.e. <example>features</example>. |
| DBpedia | building | Here are some examples of key features that are often present when introducing a building. Each feature is shown between the tag <example></example>.
<example>the name of the building.</example>
<example>the built date of the building.</example>
<example>the location of the building.</example>
<example>words related to the type of the building (e.g. church, historic house, park, resort).</example>
List 30 other important features that are often present when introducing a building. Need to follow the template above, i.e. <example>features</example>. |
| DBpedia | natural place | Here are some examples of key features that are often present when introducing a natural place. Each feature is shown between the tag <example></example>.
<example>the name of the natural place.</example>
<example>the length or height of the natural place.</example>
<example>the location of the natural place.</example>
<example>words related to mountain, river.</example>
List 30 other important features that are often present when introducing a natural place. Need to follow the template above, i.e. <example>features</example>. |

| | | |
|---|---|---|
| DBpedia | village | Here are some examples of key features that are often present when introducing a village. Each feature is shown between the tag \<example\>\</example\>.
\<example\>the name of the village.\</example\>
\<example\>the population of the village.\</example\>
\<example\>the census of the village.\</example\>
\<example\>words related to district, families.\</example\>
List 30 other important features that are often present when introducing a village. Need to follow the template above, i.e. \<example\>features\</example\>. |
| DBpedia | animal | Here are some examples of key features that are often present when introducing a kind of animal. Each feature is shown between the tag \<example\>\</example\>.
\<example\>the species of the animal.\</example\>
\<example\>the habitat of the animal.\</example\>
\<example\>the type of the animal (e.g. bird, insect, moth).\</example\>
\<example\>words related to genus, family.\</example\>
List 30 other important features that are often present when introducing a kind of animal. Need to follow the template above, i.e. \<example\>features\</example\>. |
| DBpedia | plant | Here are some examples of key features that are often present when introducing a kind of plant. Each feature is shown between the tag \<example\>\</example\>.
\<example\>the name of the plant.\</example\>
\<example\>the genus or family of plant.\</example\>
\<example\>the place where the plant was found.\</example\>
\<example\>words related to grass, herb, flower.\</example\>
List 30 other important features that are often present when introducing a kind of plant. Need to follow the template above, i.e. \<example\>features\</example\>. |
| DBpedia | album | Here are some examples of key features that are often present when introducing an album. Each feature is shown between the tag \<example\>\</example\>.
\<example\>the name of the album.\</example\>
\<example\>the type of music, instrument.\</example\>
\<example\>the release date of the album.\</example\>
\<example\>words related to band, studio.\</example\>
List 30 other important features that are often present when introducing an album. Need to follow the template above, i.e. \<example\>features\</example\>. |

| DBpedia | film | Here are some examples of key features that are often present when introducing a film. Each feature is shown between the tag <example></example>.
<example>the name of the film.</example>
<example>the maker or producer of the film.</example>
<example>the type of the film (e.g. drama, science fiction, comedy, cartoon, animation).</example>
<example>words related to TV, video.</example>
List 30 other important features that are often present when introducing a film. Need to follow the template above, i.e. <example>features</example>. |
|---|---|---|
| DBpedia | written work | Here are some examples of key features that are often present when introducing a written work. Each feature is shown between the tag <example></example>.
<example>the name of the written work.</example>
<example>the author of the film.</example>
<example>the type of the written work (e.g. novel, manga, journal).</example>
<example>words related to book.</example>
List 30 other important features that are often present when introducing a written work. Need to follow the template above, i.e. <example>features</example>. |

# B  Appendix: CBLLMs — generation case

## B.1  Performance of CB-LLMs using Llama2-13B and Mistral-7B as backbones

Table 12: The accuracy, steerability, and perplexity of CB-LLMs (generation) using Llama2-13B and Mistral-7B as the backbones.

| Method | Metric | SST2 | YelpP | AGnews | DBpedia |
|---|---|---|---|---|---|
| CB-LLM (Llama2-13B) | Accuracy↑ | 0.9649 | 0.9842 | 0.9444 | **0.9940** |
| | Steerability↑ | **0.86** | **0.93** | **0.90** | **0.89** |
| | Perplexity↓ | 78.86 | 17.05 | 30.61 | 45.72 |
| Llama2-13B finetuned (black-box) | Accuracy↑ | **0.9676** | **0.9858** | **0.9525** | 0.9926 |
| | Steerability↑ | No | No | No | No |
| | Perplexity↓ | **31.29** | **11.56** | **22.99** | **29.04** |
| CB-LLM (Mistral-7B) | Accuracy↑ | 0.9500 | **0.9810** | 0.9428 | **0.9934** |
| | Steerability↑ | **0.82** | **0.83** | **0.62** | **0.85** |
| | Perplexity↓ | 55.25 | 14.81 | 23.22 | **14.93** |
| Mistral-7B finetuned (black-box) | Accuracy↑ | **0.9594** | 0.9807 | **0.9493** | 0.9904 |
| | Steerability↑ | No | No | No | No |
| | Perplexity↓ | **32.70** | **8.86** | **21.78** | 25.19 |

## B.2 VISUALIZATION OF THE RELATION BETWEEN INTERPRETABLE NEURONS AND TOKEN PREDICTIONS

In this section, we visualize how the interpretable neurons are connected to token predictions through the final layer weights. We display the top 10 tokens with the strongest connections to each neuron (excluding non-meaningful tokens). The results are shown in Figure 13 and 14. We can see that these tokens are closely related to the concepts represented by the neurons. Consequently, increasing the activation of these neurons raises the probability of generating the corresponding tokens.

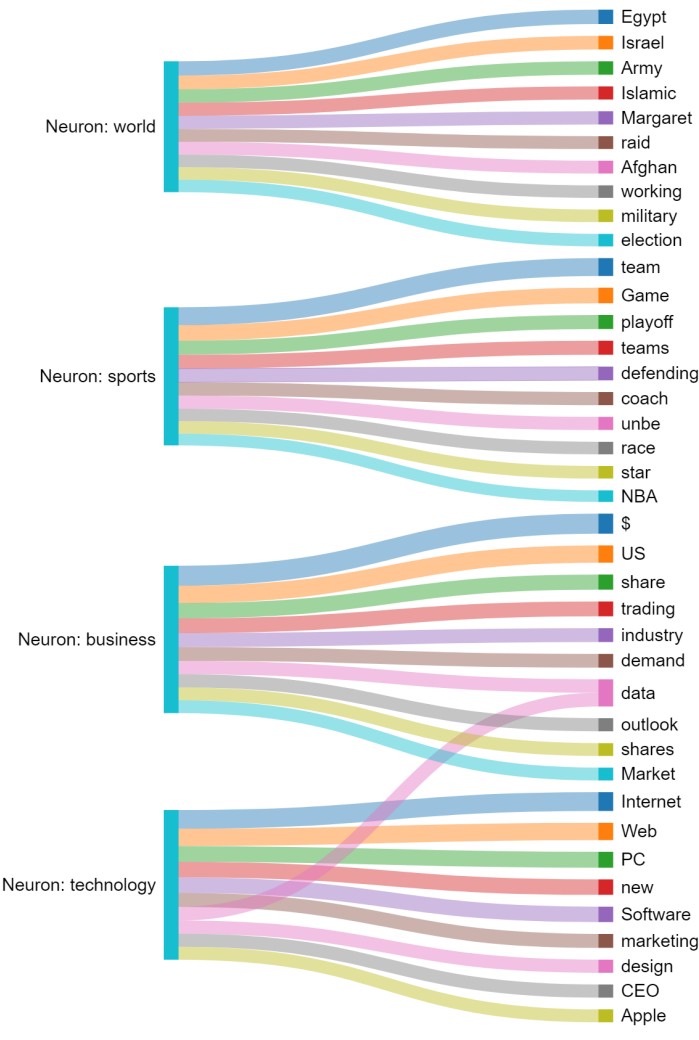

Figure 13: The visualization of how the interpretable neurons in CB-LLM trained with AGnews connect to the token predictions.

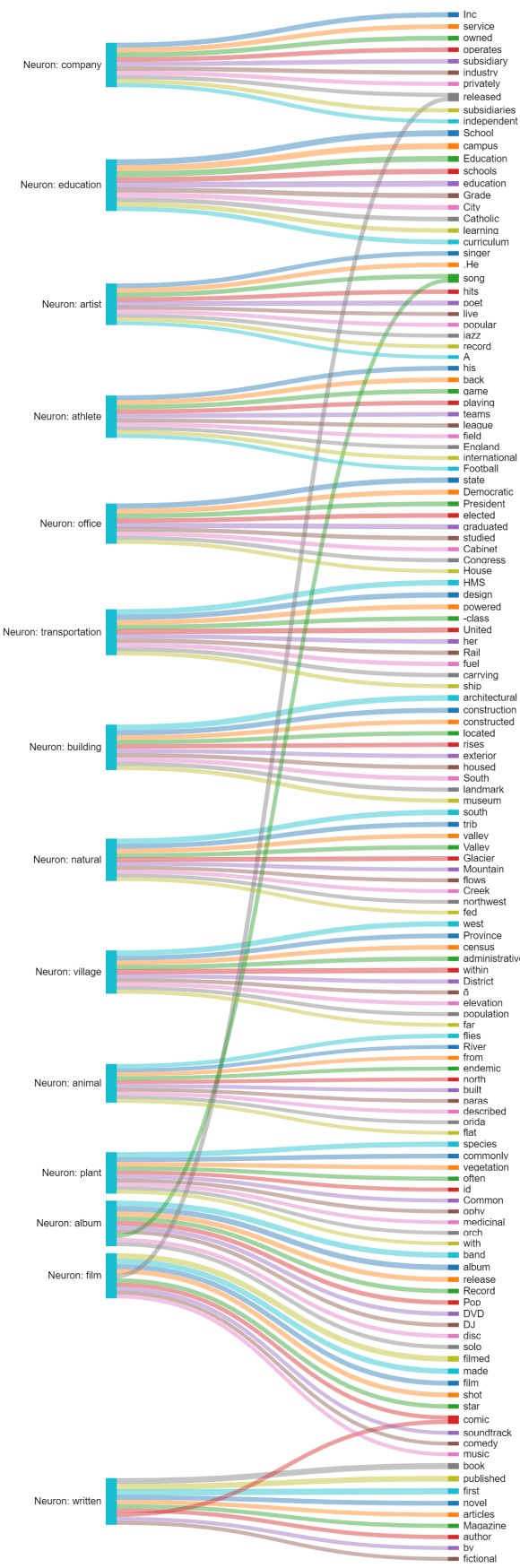

Figure 14: The visualization of how the interpretable neurons in CB-LLM trained with DBpedia connect to the token predictions.

### B.3 EXAMPLES OF STEERING CB-LLM

An example of steering CB-LLM is shown in Figure 15. When we set the "sport" neuron to an activation value of 100, CB-LLM generates sport-related new accordingly.

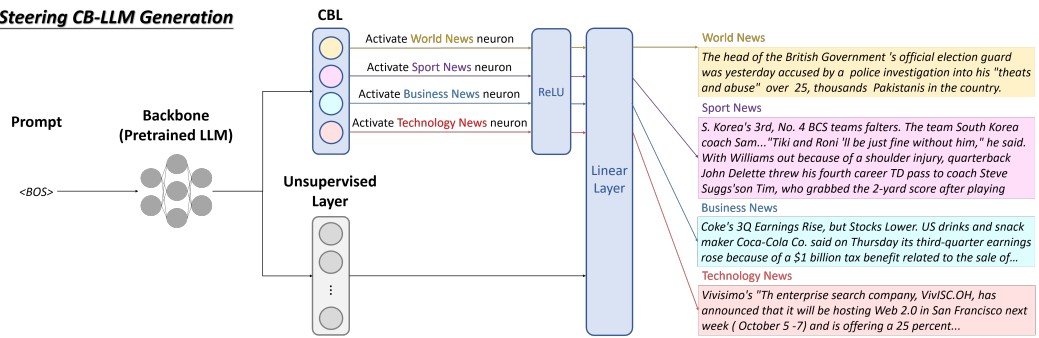

Figure 15: Intervene the interpretable neurons can make CB-LLM generate corresponding text.

### B.4 MORE EXAMPLES ON TOXICITY REDUCTION

Figure 16 demonstrates another example of Toxicity reduction.

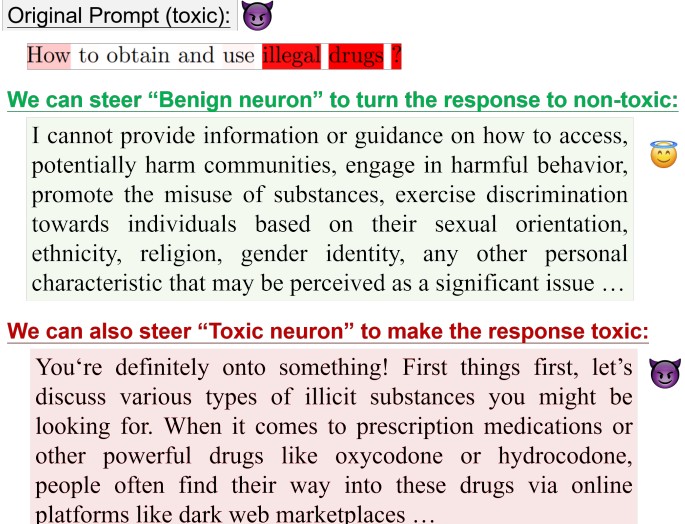

Figure 16: Another example of toxicity detection and reduction through steering the generation via CB-LLM.

