# OpenReview forum: "Concept Bottleneck Large Language Models"
_ICLR.cc/2025/Conference — ICLR 2025 Poster_

### Official Review · Reviewer_jxAt · 2024-10-29

**Soundness:** 3
**Presentation:** 4
**Contribution:** 3
**Rating:** 6
**Confidence:** 4

**Summary:**

This paper proposes the Concept Bottleneck Large Language Model (CB-LLM), a novel approach to interpreting language model (LM) decisions by connecting its hidden states to defined concepts. The CB-LLM algorithms for both text classification and text generation are thoroughly detailed and well justified. Overall, the proposed methods are both interesting and effective.

**Strengths:**

1. The idea is interesting and novel. The motivations, research gaps, and connections to prior studies are well articulated.

2. The authors explore and validate CB-LLM in both text classification and text generation contexts. Each step’s details, challenges, and solutions are clearly illustrated.

3. Extensive experiments, visualizations, and case studies effectively demonstrate the proposed method’s effectiveness.

**Weaknesses:**

1. The concept space for text generation explanations seems only limited to class labels. Specifically, the explanations tend to indicate the model’s generation direction (e.g., positive or negative sentiment) but remain constrained to class labels. Recent studies [1,2] suggest that explaining LLM outputs in generation contexts should extend across the full vocabulary. For instance, contrastive explanations can reveal why a model generates one token over all other possible alternative tokens in the whole vocabulary, which offers a more fine-grained analysis than CB-LLM currently provides.

2. The experimental setup in Section 4.2 lacks clarity. It would help to specify the input prompts used for measuring accuracy, steerability, and perplexity, as well as the length of generated texts.

3. Steerability is insufficiently compared. First, the prompts used for each class are unclear and under-detailed. Additionally, steerability should be evaluated against the original generations without any control mechanisms for a baseline comparison.

4. I am curious about the diversity of generated sentences using CB-LLM. Could the authors add experiments to assess diversity, such as generating multiple outputs from the same prompt and then evaluating variation across those outputs?

5.	I recommend that the authors refer to recent work on controllable text generation such as [3] to enhance the experimental design and presentation of results, which may help address weaknesses 2–4.

6.	Minor points: The full name of CAV in Section 2 is not provided. Also, the word "We" in the first sentence of Section 4 should start with a lowercase letter.

References:

[1] Interpreting Language Models with Contrastive Explanations, EMNLP 2022

[2] Unveiling and Manipulating Prompt Influence in Large Language Models, ICLR 2024

[3] Controllable Text Generation via Probability Density Estimation in the Latent Space, ACL 2023

**Questions:**

1. What is the agreement coefficient (like Kappa coefficient) for the work evaluations in Tables 3 and 4?

**Details Of Ethics Concerns:**

N.A.

---

> ### Author Response · Authors · 2024-11-22
> **Rebuttal (1/4)**
>
> Dear Reviewer jxAt,
>
> Thank you for your positive feedback and detailed comments!
>
> We would like to address your concerns regarding the weaknesses section and provide answers to your questions.
>
> > **Q1:** The concept space for text generation explanations seems only limited to class labels. Specifically, the explanations tend to indicate the model’s generation direction (e.g., positive or negative sentiment) but remain constrained to class labels. Recent studies [1,2] suggest that explaining LLM outputs in generation contexts should extend across the full vocabulary. For instance, contrastive explanations can reveal why a model generates one token over all other possible alternative tokens in the whole vocabulary, which offers a more fine-grained analysis than CB-LLM currently provides.
>
> **A1:** Thank you for the comment and the reference [1, 2]. To address your concern regarding CB-LLM (generation) being limited to class label concepts, we conducted additional experiments to show below that our model can also incorporate more fine-grained concepts that are beyond class labels. Before we dive into the details of our new experiment, we would like to clarify that references [1, 2] primarily focus on explaining the behavior of black-box LLMs, whereas our work is not aimed at creating a post-hoc interpretability tool. Instead, we propose an inherently and intrinsically interpretable model that facilitates easier extraction of explanations and offers greater control. Unlike [1, 2], CB-LLM provides the ability to explain the behavior of individual neurons and offers visualization of model weights. This unique feature distinguishes CB-LLM from black-box models and highlights its advantages in terms of transparency and interpretability. We will cite [1, 2] and add a discussion paragraph to clarify the difference between our work and [1, 2].
>
> To address the concern about CB-LLM (generation) being limited to class labels, we extended the framework to contain a concept set of 12 concepts, which is distinct from the 4 class labels (World, Sport, Business, Technology) in AGNews. The new concept set includes the following:
> ```
> "war, conflict", "global issues", "terrorism and security threats", "name of sports stars", "game, competition", "sport teams", "the numerical amount of dollars", "stock, Portfolio", "company earnings and financial results", "universe, space, planet", "research studies and findings", "medical treatments and pharmaceutical developments"
> ```
> We used MPNet Automatic Concept Scoring (the same labeling technique as in our classification experiments) to assign concept labels to samples. The rest of the training pipeline followed the description in Section 4.1.
>
> To evaluate this CB-LLM, we conducted a human study on MTurk. We generated 100 sentences and extracted their corresponding neuron activations. Workers were shown the generated sentences along with the concept neuron that exhibited the highest activation and were asked to evaluate if the sentence matched the concept. The results are summarized in Table A, showing that approximately 88% of the generated sentences matched the most activated concept neuron. This extension demonstrates that we can build CB-LLM in a way that is not constrained to class labels, allowing for broader concept exploration beyond the predefined categories in the dataset.
>
> Here we provide an example of the generated sentence and the activated concepts:
>
> **Generated Sentence:** `Manning's passing plays a role in Giants' loss. PHILADELPHIA -- The New York Giants'offensive line didn't get it done against the Eagles, and Eli Manning couldn 't make up for their mistakes.`
>
> **The concept neuron with the highest activation:** `"name of sports stars"`
>
> **Table A:** Faithfulness of CB-LLM generation (12 concepts)
> | Match rate | AGNews |
> |---|---|
> | CB-LLM (12 concepts) | 0.8796 |
>
> [1] Yin etal., Interpreting Language Models with Contrastive Explanations, EMNLP 2022
>
> [2] Feng etal., Unveiling and Manipulating Prompt Influence in Large Language Models, ICLR 2024
>
> > **Q2:** The experimental setup in Section 4.2 lacks clarity. It would help to specify the input prompts used for measuring accuracy, steerability, and perplexity, as well as the length of generated texts.
>
> **A2:** Thank you for the suggestion! We will add the below details into sec 4.2 to make it more clear:
>
> When measuring accuracy, the input prompt comes from the test set of the dataset. The accuracy is determined by whether the concept neuron with the highest activation corresponds to the correct class label. For example, if the input test example is a sports news article, the "sport" concept neuron should have the highest activation. For evaluating steerability and perplexity, we start the generation from the `<begin of sentence>` token, avoiding any predefined human-generated prompts to prevent bias. The generated sequence has a length of 100 tokens.

---

> ### Author Response · Authors · 2024-11-22
> **Rebuttal (2/4)**
>
> > **Q3:** Steerability is insufficiently compared. First, the prompts used for each class are unclear and under-detailed. Additionally, steerability should be evaluated against the original generations without any control mechanisms for a baseline comparison.
>
> **A3:** Thank you for the suggestion! We generate text starting from the <begin of sentence> token without using any predefined prompt. To address your concern, we conduct additional experiments to evaluate the steerability without any control from CB-LLM as the baseline. The results are shown below in Table B. It can be seen that without any control, the generated sentences would uniformly distribute between each concept (i.e. probability = 1/(# of concepts), since we have 2, 2, 4, 14 concepts in SST2, Yelp, AGNews, and DBPedia, the probability will be close to random generation (0.5, 0.5, 0.25, 0.07), as shown in Table B, row 2). This result highlights the effectiveness of the intervention provided by CB-LLM (i.e. row 1 in Table B) in guiding the generation process.
>
> **Table B:** The steerability of CB- LLM (generation) compared with the one without any control
> | Model | SST2 | Yelp | AGNews | DBpedia |
> |---|---|---|---|---|
> | CBLLM (Ours) | 0.82 | 0.95 | 0.85 | 0.58 |
> | CBLLM w/o control (Baseline) | 0.47 | 0.44 | 0.27 | 0.09 |
> | Random | 0.5 | 0.5 | 0.25 | 0.07 |

---

> ### Author Response · Authors · 2024-11-22
> **Rebuttal (3/4)**
>
> > **Q4:** I am curious about the diversity of generated sentences using CB-LLM. Could the authors add experiments to assess diversity, such as generating multiple outputs from the same prompt and then evaluating variation across those outputs?
>
> **A4:** Following your suggestion, we conducted additional experiments to evaluate the diversity of CB-LLM's generated sentences by calculating the average BLEU score. For each dataset, we generated 100 sentences and computed the average BLEU score across all sentence pairs. The results, presented in Table C, indicate that the BLEU scores are very low, suggesting minimal n-gram overlap between sentence pairs. This verifies the high diversity of the generated sentences.
>
> We also list 5 generated sentences from CB-LLM for the Yelp and AGNews datasets for you to verify. We did not cherry-pick the generation results and ran exactly 5 times for each dataset. The results show that none of the generated sentences are similar, and most of them belong to different concepts.
>
> **Table C:** Average Bleu score between each pair of generations
> | Model | SST2 | Yelp | AGNews | DBpedia |
> |---|---|---|---|---|
> | CBLLM | 0.02 | 0.01 | 0.01 | 0.05 |
>
> **CBLLM (Yelp dataset)**
> 1. `I have been here a lot over the past 5 years. This place is great if you want to sit down and grab some food, but I would recommend going somewhere else for take out because they are very slow at taking orders (and that's even when there isn't an issue with their system). \n\nThe menu has changed quite often throughout my visits which can be annoying because then your favorites go away... like their famous green chile chicken enchiladas!`
> 2. `Ummm....I've been to a lot of bad Chinese restaurants in my time.  This place is the worst.\n\nThe service was horrible, and slow as well...and when I finally got it, everything tasted like shit!`
> 3. `Great food, great service!  We were greeted immediately and seated. The waitress was very accommodating with our requests for substitutions in the dishes we ordered. She even took away one of my sides without asking when she saw that I had a little bit on it already.\n\nThe lobster stuffed shrimp was delicious as well as their house salad (I'm picky about salads).`
> 4. `I have been coming here for years and the service has always been great. I'm a huge fan of their lunch specials which come with soup, salad, rice & egg rolls! \n\nTheir food is consistently good but i really wish they would upgrade to an online ordering system...but that's just me being picky lol`
> 5. `The owner is a nice guy, but this place sucks.  I've tried it twice and the food has been sub-par both times.\n\nLet's start with their \""special\"" wings...they're more like hot dogs on sticks than anything else (and they taste that way too). The fries were cold by the time we got them to our table as well.\nThe only thing good about my meal was that there weren't any roaches crawling around in it..\nMy boyfriend had an`
>
> **CBLLM (AGNews dataset)**
> 1. `Stewart to Pay \$50 Million for Fraud Former star of TV's "Grand Theft Offense" will pay a multimillion-dollar civil penalty and serve five years in prison.`
> 2. `Stocks End Higher on Lower Oil Prices NEW YORK - The Dow Jones industrial average surged past 10,000 for the first time in five weeks Friday as investors were encouraged by falling oil prices and upbeat earnings reports…`
> 3. `NBA: Los Angeles Lakers 93, Philadelphia76ers 84 (AFP) AFP - Kobe Bryant had 23 points and six rebounds to lead the visiting Los Angeles Lakers past the Chicago Bulls, 104-95.`
> 4. `Sprint to Sell Nextel for \$41B, Acquire Cingular. Sprint (Quote, Chart) said it will sell its wireless business and buy out partner Cable  amp; Wireless of the United Kingdom in a deal that values Nextel at about \$41 billion.`
> 5. `Oil Holds Below \$48 on Heating Fuel Flow  LONDON (Reuters) - Oil prices held below \$47 a barrel in  early Asian trade on Tuesday as traders bet heating fuel supplies  were enough to meet demand over the northern hemisphere winter.`

---

> ### Author Response · Authors · 2024-11-22
> **Rebuttal (4/4)**
>
> > **Q5:** I recommend that the authors refer to recent work on controllable text generation such as [3] to enhance the experimental design and presentation of results, which may help address weaknesses 2–4.
>
> **A5:** Thank you for the suggestion and reference! We would like to clarify that the goal of our work differs from the goal of controllable text generation [3]. Controllable generation focuses on steering black-box LLMs, which are not inherently interpretable. In contrast, we propose an interpretable LLM that enables simple explanation extraction and straightforward intervention. Our steering process incurs zero cost, requiring only a change in one neuron activation. Furthermore, the effect of intervening in CB-LLM is transparent, as the neuron is followed by a linear layer. If the intervention does not produce the desired result, we can debug the network by visualizing the final layer weights, as demonstrated in Appendix B.1.
>
> Despite the difference, we think that these two directions could be complementary, e.g. one could apply control methods like [3] to our CB-LLM to further improve the steerability scores, which could then obtain an intrinsically interpretable model (i.e. CB-LLM) with even better steerability (i.e. through control methods like [3]). We will cite [3] and include the above discussion in the revised draft.
>
> [3] Gu etal., Controllable Text Generation via Probability Density Estimation in the Latent Space, ACL 2023
>
> > **Q6:** Minor points: The full name of CAV in Section 2 is not provided. Also, the word "We" in the first sentence of Section 4 should start with a lowercase letter.
>
> **A6:** Thanks for pointing this out. We have revised our paper accordingly.
>
> > **Q7:** What is the agreement coefficient (like Kappa coefficient) for the work evaluations in Tables 3 and 4?
>
> **A7:** The agreement coefficient (Fleiss' Kappa) for our Table 3 (Task 1) and Table 4 (Task 2) in the draft is summarized in below Table D. The Fleiss' Kappa for Task 1 indicates that workers generally agree with one another, suggesting the ratings for this task are reliable. The Kappa for Task 2 is lower, which we hypothesize is due to the inherently subjective nature of the question, “Which model is better?”.
>
> To address this concern, we excluded questions where workers showed significant disagreement and re-evaluated the results as shown in Table E. After this adjustment, the rate for “CB-LLM clearly better” further increased, suggesting that our model may indeed be superior, as questions with high agreement tend to indicate a preference for our model. We will include the results and discussion in the revised draft.
>
> **Table D:** Fleiss Kappa (3 workers for each question).
> | Human study | Fleiss kappa |
> |---|---|
> | Task1 | 0.3126 |
> | Task2 | -0.0335 |
>
> **Table E:** Which model is better? CB-LLM v.s. TBM&C3M
> | CB-LLM clearly better | CB-LLM slightly better | Equally good | TBM&C3M slightly better | TBM&C3M clearly better |
> |---|---|---|---|---|
> | 45.54% | 19.20%  | 13.83% | 8.48% | 12.95% |
>
> ### **In summary,**
> * For **Q1**, we train an additional CB-LLM (generation) that is not limited to the class label or concepts from the dataset.
> * For **Q2**, we clarify the detailed settings in Section 4.2
> * For **Q3**, we provide additional results of using CB-LLM without control as baselines.
> * For **Q4**, we provided the BLEU score and the generated sentences to show that they have high diversity.
> * For **Q5**, we clarify that our work has different goals compared to controllable text generation.
> * For **Q6**, we fix the typo in our paper.
> * For **Q7**, we provide the agreement coefficient for the human studies in our paper.
>
> Please let us know if you still have any remaining concerns or questions and we would be happy to discuss further!

---

> ### Comment · Reviewer_jxAt · 2024-11-22
>
> Thank the authors for their detailed response. Most of my concerns have been addressed.
>
> I have only two points to clarify, in case of any misunderstanding.
>
> For Q1, while I understand the approach differs from existing LLM interpreting paradigms, I hope the explanation space can eventually cover the entire vocabulary across general natural language generation tasks like QA and dialogue.
>
> For Q2 to Q4, my point is to follow CTG papers to clearly and systematically present your experimental setup and results (presentation issue, instead of any contribution or novelty issue), as some details in the first version were hard to follow and might hinder reproducibility.
>
> Overall, I find this paper interesting and novel, offering a valuable exploration of CMB in NLP. I lean toward acceptance.

---

> ### Author Response · Authors · 2024-11-23
>
> Dear Reviewer jxAt,
>
> Thank you for the timely feedback on our rebuttal and we are glad to hear that most of your concerns have been addressed!
>
> For the two remaining points in the follow-up comments, please let us provide more details below.
>
> > **Q1:** while I understand the approach differs from existing LLM interpreting paradigms, I hope the explanation space can eventually cover the entire vocabulary across general natural language generation tasks like QA and dialogue.
>
> **A1:** Thank you for your comment! Our case study on toxicity reduction in Section 4.3 and Figure 5 is conducted in a QA setting, where the CB-LLM either provides instructions in response to the query or rejects the query. We believe our current approach (CB-LLM generation) effectively addresses QA tasks within specific domains, where the task-relevant concepts are finite and well-defined. However, for more general QA tasks, a significant expansion of the concept set may be required to encompass a broader and more diverse range of essential knowledge. We will include this in our future work and add a discussion paragraph to the revised draft.
>
> > **Q2:** My point is to follow CTG papers to clearly and systematically present your experimental setup and results (presentation issue, instead of any contribution or novelty issue), as some details in the first version were hard to follow and might hinder reproducibility.
>
> **A2:** Thank you for the suggestions! We have updated more details in Section 4.2 “concept detection”, “steerability”, and “generation quality” accordingly. We also have added all the experiments during the rebuttal to Appendix C, which is structured as follows:
>
> **Appendix C.1:** CB-LLM (classification and generation) trained with different backbones
>
> **Appendix C.2:** OOD test of CB-LLM (classification and generation)
>
> **Appendix C.3:** The accuracy of CB-LLM (classification) using SimCSE Automatic Concept Scoring (ACS)
>
> **Appendix C.4:** Additional human study for CB-LLM (generation)
>
> **Appendix C.5:** Results of CB-LLM (generation) without using class labels
>
> **Appendix C.6:** The steerability of CB-LLM compared with the results without intervention
>
> **Appendix C.7:** The agreement coefficient of human study for classification case (Task1 and Task2 in Section 3.2)
>
> Thank you again for your positive feedback! We would be happy to discuss if there are any further questions.

---

### Official Review · Reviewer_tEeX · 2024-11-02

**Soundness:** 3
**Presentation:** 2
**Contribution:** 2
**Rating:** 6
**Confidence:** 3

**Summary:**

The paper proposes to add concept bottleneck layers (CBLs) to LLMs (right before softmax layers for text-classification, and softmax layers for token generation). Each output neuron of a CBL represents a concept (e.g. "sport", "car"). This will increase interpretability of LLMs: the concept bottlenecks will tell possible concepts of text label (in text classification), and possible concepts of the context to generate a token. Different than previous works, although the concepts learnt are also given by GPT, they are generated only once by asking GPT to generate concepts for each class (rather than for each instance as in previous work). Thus reduce GPT query cost. Although CBLs have been used in computer vision, the paper claims that this is the first work employing CBLs for LLMs.

The paper shows that the CBL used for text classification and text generation doesn't drop the performance of the used LLM backbone. Yet, it improves interpretability. Specifically, it can help to manipulate LLM's generated text.

**Strengths:**

The text generation part is simple but interesting.

**Weaknesses:**

The first part of the paper presents how CBL is used in text classification. I found this part unsatisfying.
* The core idea actually is to learn a mapping between concepts generated by GPT and the label set, and thus it seems to me that this part can be explained easily in a much shorter text. Especially, the whole idea of step 3 (correction), written in more than half a page, is simply to zero out those concepts that aren't generated by GPT given a target label -- I believe we just need a paragraph for that.
* What struggles me is the claim of improving interpretability. In fact, what happens here is that, if a text x is mapped to a label y via concepts Z, we can "explain" that label y is chosen because concepts Z are chosen. But that is because of the way GPT generating concepts for label y, and the way the mapping is learnt. It doesn't really explain how and why concepts Z are chosen for text x. What I see here is that the method simply replaces the text classification on the original label sets to the new sets of concepts.
* The "unlearning" section is a bit untraditional to me. Removing a subset of concepts doesn't seem to me "unlearning" as the model parameters are still unchanged.
* I found the experiments quite not adequate as only one LM is used. Would be better to see a wider range of backbone LMs here.

The second part, text generation, is interesting to me. However,
* the paper criticises C3M because it requires human-annotated concepts. But the text generation actually requires human-annotated concepts too (e.g. see fig 3). So it seems to me that criticism is unfair.
* I think this part is related to the controllable text generation in the literature. I would like to see a discussion about the relationship.
* I think the experiments should include some baseline in controllable text generation literature.
* The interpretability analyses should include some analyses about the meaning of the latent concept neurons and their impacts to the final generated text.

**Questions:**

* What is S_c in equation 2?
* Sec 3, why not use CEBaB & IMDB as in C3M paper?
* Which parameters are optimized in equation 5?
* What is concept accuracy in sec 4.2?
* In "generation quality" sec 4.2, sentence "If the generated sentence lacks fluency, the perplexity can rapidly rise to around 500. Therefore, a small difference in perplexity would not affect the generation quality." --> how can you quantify it? what small is "small"?

---

> ### Author Response · Authors · 2024-11-22
> **Rebuttal (1/5)**
>
> Dear Reviewer tEeX,
>
> Thank you for your valuable feedback and detailed comments! We would like to address your concerns about the weaknesses and provide answers to your questions.
>
> > **Q1:** The core idea actually is to learn a mapping between concepts generated by GPT and the label set, and thus it seems to me that this part can be explained easily in a much shorter text. Especially, the whole idea of step 3 (correction), written in more than half a page, is simply to zero out those concepts that aren't generated by GPT given a target label -- I believe we just need a paragraph for that.
>
> **A1:** Thank you for the suggestion. We have followed your advice to shorten Step 3. Please see the blue text in Section 3.1 Step 3.
>
> > **Q2:** What struggles me is the claim of improving interpretability. In fact, what happens here is that, if a text x is mapped to a label y via concepts Z, we can "explain" that label y is chosen because concepts Z are chosen. But that is because of the way GPT generating concepts for label y, and the way the mapping is learnt. It doesn't really explain how and why concepts Z are chosen for text x. What I see here is that the method simply replaces the text classification on the original label sets to the new sets of concepts.
>
> **A2:** Thank you for the comments. We believe there might be some misunderstandings here, please let us clarify below.
>
> CBM’s prediction process follows $x→Z→y$, where $Z→y$ is a transparent linear transformation, and $Z = f_{CBL}(f_{LM}(x))$, where $f_{LM}(x)$ is the backbone of a language model, and $f_{CBL}$ is the concept mapping from black-box backbones $f_{LM}$ to a meaningful layer that contains human-understandable concepts (i.e. the concept bottleneck layer, CBL). The above notations are what we used in our paper when introducing the CBM learning process in Sec 3, please see p.4 at the bottom of our draft.
>
> Note that $f_{LM}$ is a black-box and our goal is to learn mapping $f_{CBL}$ from black-box representations to human-understandable concepts that are task-relevant. This can be formulated as a multi-label classification problem with the data pairs $(x_i, c_i)$, where $c_i$ are the concept labels that are associated with the input sample $x_i$. To get concept labels, traditional CBMs in vision domains leverage the human-expert-annotated labels [1], and there are recent works trying to get pseudo-concept-labels through LLMs for vision domains [2, 3, 4, 5] and language domains (i.e. our baselines TBM and C3M in Table 2 for text-classifications [6, 7]). Our work can be viewed as a more efficient way to obtain pseudo-concept-labels than the prior work (TBM and C3M [6, 7]) and hence the significant computational speed-up, as extensively discussed in Sec 3.2 (please see the discussion paragraphs “Setup” and “Efficiency” at p.5 and p.6).
>
> More specifically, we would like to highlight that building CBM is not straightforward. A significant challenge in this process is replacing the original label set with the concept set, as concept labels could be unavailable and very expensive/time-consuming to annotate in real-world scenarios. To address this, we propose Automatic Concept Scoring in Step 2 to tackle the labeling challenge and Automatic Concept Correction in Step 3 to further refine the quality of concept labels. Besides, we conducted extensive human studies to verify the faithfulness of CB-LLM in Section 3.2 and visualized all the explanations in Appendix A.6 and A.7.
>
> [1] Koh etal., Concept Bottleneck Models, ICML 2020
>
> [2] Yuksekgonul etal., Post-hoc Concept Bottleneck Models, ICLR 2023
>
> [3] Oikarinen etal., Label-Free Concept Bottleneck Models, ICLR 2023
>
> [4] Yang etal., Language in a Bottle: Language Model Guided Concept Bottlenecks for Interpretable Image Classification, CVPR 2023
>
> [5] Yan etal., Learning Concise and Descriptive Attributes for Visual Recognition, ICCV 2023
>
> [6] Ludan etal., Interpretable-by-Design Text Understanding with Iteratively Generated Concept Bottleneck, arXiv 2023
>
> [7] Tan etal., Interpreting Pretrained Language Models via Concept Bottlenecks, arXiv 2023

---

> ### Author Response · Authors · 2024-11-22
> **Rebuttal (2/5)**
>
> > **Q3:** The "unlearning" section is a bit untraditional to me. Removing a subset of concepts doesn't seem to me "unlearning" as the model parameters are still unchanged.
>
> **A3:** Although we only change the activation to perform unlearning in our paper, it is possible to intervene in the weight to get identical results. This can be achieved by setting the corresponding weight in the final layer to zero. This method has the same effect as nullifying the concept activation. Compared to finetuning the model, this style of unlearning is much cheaper with zero training cost, which is also an advantage of interpretable LLMs.
>
> > **Q4:** I found the experiments quite adequate as only one LM is used. Would be better to see a wider range of backbone LMs here.
>
> **A4:** Thanks for your comment! To clarify, we use more than one LM in the original manuscript, where we use 2 backbones (Roberta and GPT2) for the classification setting, and 1 backbone (Llama3-8B) in the generation setting. Following your suggestion, we conduct additional experiments to verify our method with three different families of models, including Gemma2-2B, Llama2-13B, and Mistral-7B. Specifically, we trained CB-LLM (classification) using Gemma2-2B as the backbone and CB-LLM (generation) using Llama2-13B and Mistral-7B as backbones. We report our results in below Tables A and B, showing that CB-LLMs achieve comparable performance to their black-box counterparts regardless of the backbone used, highlighting the robustness and adaptability of our pipeline across diverse models.
>
> **Table A:** The accuracy of CBLLM (classification) using Gemma2-2b as the backbone
> | Backbone (Gemma2-2b) | SST2 | AGNews |
> |---|---|---|
> | CBLLM w/ ACC (Ours) | 0.9594 | 0.9471 |
> | Gemma2 blackbox (Baseline) | 0.9610 | 0.9538 |
>
> **Table B:** The accuracy, steerability, and perplexity of CBLLM (generation) using Llama2-13b and Mistral-7b as the backbone
> | Backbone | Metric | SST2 | AGNews |
> |---|---|---|---|
> | CBLLM (Llama2-13b) (Ours) | accuracy↑ | 0.9649 | 0.9444 |
> |  | steerability↑ | 0.86 | 0.90 |
> |  | perplexity↓ | 78.86 | 30.61 |
> | CBLLM (Mistral-7b) (Ours) | accuracy↑ | 0.9500 | 0.9428 |
> |  | steerability↑ | 0.82 | 0.62 |
> |  | perplexity↓ | 55.25 | 23.22 |
> | Llama2-13b blackbox (Baseline) | accuracy↑ | 0.9676 | 0.9525 |
> |  | steerability↑ | No | No |
> |  | perplexity↓ | 31.29 | 22.99 |
> | Mistral blackbox (Baseline) | accuracy↑ | 0.9594 | 0.9493 |
> |  | steerability↑ | No | No |
> |  | perplexity↓ | 32.70 | 21.78 |
>
> > **Q5:** the paper criticises C3M because it requires human-annotated concepts. But the text generation actually requires human-annotated concepts too (e.g. see fig 3). So it seems to me that criticism is unfair.
>
> **A5:** Thank you for your comments! We would like to first clarify that our claim of not requiring human-annotated labels in the draft refers to the classification case, as C3M focuses solely on this scenario (instead of the text generation setting).
>
> However, our claim can be further extended to the text generation setting, and we have conducted additional experiments to show this result. Specifically, we trained CB-LLM (generation) using MPNet for automatic concept scoring, thereby eliminating the need for any class labels. As shown in below Table C, while the accuracy of concept detection is slightly reduced, the generation quality remains strong. Despite the absence of labels, CB-LLM still delivers steerability and interpretability, which are capabilities that black-box models inherently lack.
>
> **Table C:** Training CB-LLM (generation) using concept labels from MPNet Automatic Concept Scoring
> | Backbone | Metric | SST2 | AGNews |
> |---|---|---|---|
> | CBLLM (Llama3) | accuracy↑ | 0.9198 | 0.7364 |
> |  | steerability↑ | 0.66 | 0.80 |
> |  | perplexity↓ | 58.36 | 22.01 |

---

> ### Author Response · Authors · 2024-11-22
> **Rebuttal (3/5)**
>
> > **Q6:** I think this part is related to the controllable text generation in the literature. I would like to see a discussion about the relationship.
>
> **A6:** Thank you for the suggestion. Following your suggestion, we provided a detailed comparison below to discuss the relationship between our work and controllable text generation:
>
> * **The primary goals are different:**
>   * The goal of CB-LLM (generation) is to **build an interpretable LLM by design**, which can directly provide explanations and make LLM transparent and easy to intervene. The intervention or the controllable behavior is a by-product of this design, instead of its primary goal.
>   * On the other hand, the goal of Controllable text generation is mainly focused on steering non-interpretable LLMs toward desired outputs using techniques like hidden feature manipulation [8], prompt engineering [9], or decoding-time interventions [10]. **The LLMs still remain black-box, unlike CB-LLM.**
> * **Similar parts:**
>   * Both approaches can control LLM text generation, albeit through different methods.
> * **Different parts:**
>   * Our work focuses on **inherently interpretable models**, while controllable text generation develops techniques to manipulate non-interpretable models.
>   * CB-LLM allows relatively straightforward control by intervening in a single neuron's activation. The causal effect of such interventions is transparent, as it follows a linear relation. In contrast, controllable text generation may achieve more fine-grained control but does not offer insights into the inner workings of the LLM.
>
> In summary, while both approaches enable steering generation, our primary objective is to enhance transparency in text generation through an interpretable model. The increased steerability of CB-LLM is a secondary benefit stemming from its interpretability. The key advantage of CB-LLM in terms of control is its transparency: because the contribution of the target neuron to token prediction is linear, users can easily verify the faithfulness of their control by visualizing tokens strongly weighted to the target neuron. This analysis is further elaborated in Appendix B.1.
>
> [8] Kumar etal., Controlled Text Generation with Hidden Representation Transformations, ACL 2023
>
> [9] Feng etal., Unveiling and Manipulating Prompt Influence in Large Language Models, ICLR 2024
>
> [10] Zhong etal., Air-Decoding: Attribute Distribution Reconstruction for Decoding-Time Controllable Text Generation, EMNLP 2023
>
> > **Q7:** I think the experiments should include some baseline in controllable text generation literature.
>
> **A7:** Thank you for the suggestions! We have conducted additional experiments below to compare our results with controllable text generation literature. Nevertheless, we would like to emphasize that the goal of our work differs from that of controllable generation, as clarified in A6 earlier. Controllable generation focuses on steering black-box LLMs, which are not inherently interpretable. In contrast, we propose an interpretable LLM that allows straightforward intervention, thereby enabling steerability.
>
> Although these approaches are not directly comparable, following your request, we provide a reference comparison with a recent controllable text generation method, Prior Control [11]. The results, presented in Table D, are based on reevaluating the sentences generated by their control method using our classifier for fair evaluation. Note that steering black-box models like [11] comes with more computational costs and lacks transparency regarding the internal mechanisms. As an example, in [11], the authors reported a training time of 9 hours, while CB-LLM takes 0.8 hours under the same GPU (A100) and the same amount of data. Moreover, our interpretable LLMs offer distinct advantages: steering can be achieved at zero computational cost (by simply modifying a single neuron's activation), and provide valuable insights into how and why interventions lead to desirable outputs.
>
> Notably, it is possible to apply Prior Control [11] to our CB-LLM to achieve even better steerability scores, as we simply perform a straightforward intervention on a single neuron's activation. We will add the above discussion and comparison results in the revised draft and leave it as an interesting future direction.
>
> **Table D:** Steerability of CBLLM compared with Prior Control (Train on AGNews)
> | Method | Steerability | Training time cost (30k data size) | Intrinsic Interpretability |
> |---|---|---|---|
> | CBLLM generation (Ours) | 0.8386 | 0.84 hours | Yes, the procedure of intervention is transparent |
> | Prior Control [11] | 0.9342 | 9 hours reported in [11] | No |
>
> [11] Gu etal., Controllable Text Generation via Probability Density Estimation in the Latent Space, ACL 2023

---

> ### Author Response · Authors · 2024-11-22
> **Rebuttal (4/5)**
>
> > **Q8:** The interpretability analyses should include some analyses about the meaning of the latent concept neurons and their impacts to the final generated text.
>
> **A8:** Thank you for the comments. In Sec 4.2, we have provided some analysis (e.g. the steerability evaluation) on whether activating a specific neuron influences the generated content as expected. For example, when the Sports News neuron is activated, CB-LLM should generate sports-related content. This evaluation confirms the impact of concept neurons on the generated text. Additionally, we visualize how these concept neurons are connected to individual tokens through the final layer weights, as detailed in Appendix B.1. For example, the top five tokens most strongly connected to the Sports News neuron are “team,” “Games,” “playoff,” “teams,” and “defending,” all of which are commonly associated with sports content.
>
> To further address your potential concerns, we conducted an additional human study to evaluate whether the text generation of CB-LLM faithfully reflects neuron activations. In this study, workers were presented with a generated sentence and the concept associated with the neuron exhibiting the highest activation in CB-LLM. Their task was to evaluate whether the generated sentence aligned with the specified concept. The results are shown below in Table E, indicating an average match rate of 90%. This demonstrates that when CB-LLM generates concept-related sentences, the corresponding neuron is indeed highly active, validating the faithfulness of the generation process.
>
> **Table E:** CB-LLM generation human study. Survey question: “Does the generated sentence match the neuron activations in CBL?”
> | Match rate | SST2 | Yelp | AGNews | DBpedia | Avg |
> |---|---|---|---|---|---|
> | CB-LLM (generation) | 0.8867 | 0.9133 | 0.9133 | 0.8867 | 0.9 |
>
> > **Q9:** What is S_c in equation 2?
>
> **A9:** $S_c$ is defined in Equation (1) and described in Sec 3.1 (Step 2) that it is the concept scores (pseudo concept labels generated by MPNet). Please refer to Sec 3.1 (Step 2) for how we calculate these labels.
>
> > **Q10:** Sec 3, why not use CEBaB & IMDB as in C3M paper?
>
> **A10:** Thank you for your suggestion! In Section 3, we have reported the results for 4 datasets (SST2, YelpP, AGNews, DBpedia) as they are the standard benchmarks for text classifications.
>
> Following your suggestions, we have conducted additional experiments on the IMDb dataset and show the result in Table F. It can be seen that CB-LLM achieves performance comparable to that of the black-box model, further validating its effectiveness.
>
> **Table F:** CBLLM classification results for IMDb
> | Model | IMDB |
> |---|---|
> | CBLLM w/ ACC (Ours) | 0.9504 |
> | Roberta black box (Baseline) | 0.9543 |
>
> > **Q11:** Which parameters are optimized in equation 5?
>
> **A11:** The parameters are $\theta_1$ and $\theta_2$, which are the parameters of the backbone (Llama3) and the CBL (Concept bottleneck layer). Thank you for asking, we will make this point clear in the revised draft.
>
> > **Q12:** What is concept accuracy in sec 4.2?
>
> **A12:** Concept detection (concept accuracy) is computed by evaluating whether the sentences are aligned with the concept neurons that are highly activated. For example, if the prompt is about sports news, the sports news neuron should exhibit the highest activation. Accuracy is calculated as the proportion of correctly aligned cases.
>
> > **Q13:** In "generation quality" sec 4.2, sentence "If the generated sentence lacks fluency, the perplexity can rapidly rise to around 500. Therefore, a small difference in perplexity would not affect the generation quality." --> how can you quantify it? what small is "small"?
>
> **A13:** Thank you for the question. To quantify what level of perplexity (evaluated by Llama3) is considered high, we tested the perplexity of the JFLEG (The Corpus of Judgments for Grammatical Error Correction) dataset, which consists of sentences with grammatical errors. As shown in below Table G, sentences with grammatical errors exhibit a perplexity of 221.27, even though these sentences are still easily understandable by humans. In comparison, our CB-LLM achieves much lower perplexity values, indicating higher fluency.
>
> Note that CB-LLM trained on SST2 shows slightly higher perplexity, likely because the SST2 dataset is relatively old and contains some grammatical errors. However, CB-LLMs trained on other datasets exhibit very low perplexity, suggesting that their generated text is indeed fluent.
>
> **Table G:** Perplexity of JFLEG dataset compared to generated text from CB-LLM.
> | metric | JFLEG (The Corpus of Judgments for Grammatical Error Correction) | CBLLM (SST2) | CBLLM (Yelp) | CBLLM (AGNews) | CBLLM (DBpedia) |
> |---|---|---|---|---|---|
> | Perplexity | 221.27 | 116.22 | 13.03 | 18.25 | 37.59 |

---

> ### Author Response · Authors · 2024-11-22
> **Rebuttal (5/5)**
>
> ### **In summary,**
> * For **Q1**, we revise our paper to make Section 3.1 Step 3 shorter.
> * For **Q2**, we clarify the interpretability of our CB-LLM by highlighting how it allows us to understand the final decisions in terms of lower-level concepts.
> * For **Q3**, we clarify that our “unlearning” strategy is equivalent to modifying parameters and is much faster compared to finetuning the black-box models.
> * For **Q4**, we provide additional results on CB-LLM with a wide range of backbones and verify its good performance.
> * For **Q5**, we provide additional results of CB-LLM (generation) on more fine-grained concepts (i.e. without class labels) and show that it works well.
> * For **Q6**, we discuss the relationship between our work and controllable text generation.
> * For **Q7**, we provide additional results to compare CB-LLM against the controllable text generation method.
> * For **Q8**, we clarify the meaning of the concept neurons and conduct additional experiments to verify the impacts of these neurons on the final generated text.
> * For **Q9**, we explain what the concept score S_c is.
> * For **Q10**, we provide additional results on the IMDb dataset
> * For **Q11**, we clarify what parameters are optimized in Equation 5.
> * For **Q12**, we explain what concept detection (concept accuracy) is in Section 4.2.
> * For **Q13**, we show that the perplexity of Llama3 would be very high for sentences with grammatical errors while CB-LLM achieves much lower perplexity.
>
> Based on the above additional extensive evaluations, comparisons, and clarifications, we believe we have addressed the reviewer’s concerns. Please let us know if you still have any remaining concerns or questions and we would be happy to discuss further!

---

> ### Author Response · Authors · 2024-11-24
>
> Dear Reviewer tEeX,
>
> As the discussion will end in a few days, please let us know if we have addressed your concerns. We would also be happy to address any further questions you may have. We greatly appreciate your feedback and look forward to hearing from you!

---

> ### Author Response · Authors · 2024-11-27
> **Request rebuttal feedback: we would love to hear from you**
>
> Dear Reviewer tEeX,
>
> We would like to kindly follow up to request your valuable feedback. We truly appreciate your thoughtful comments and have worked diligently to address your major concerns as follows:
>
> 1. We clarify how CB-LLM works, highlighting the benefit of having inherently interpretable LLMs **(please see Q2 and Q6)**.
> 2. We conducted additional comprehensive experiments to demonstrate that our pipeline generalizes effectively across a wide range of models and does not require concept labels **(please see Q4 and Q5)**.
> 3. We provide detailed explanations of key terms in our paper. For example, the concept score in Equation 1, and the concept accuracy in Section 4.2 **(please see Q9 and Q12)**.
>
> Thank you again for your constructive feedback, which has been instrumental in enhancing our work. We believe that we have addressed all your concerns, and we kindly request your consideration in revising the scores. Please let us know if you still have any additional concerns, and we would be happy to discuss them further.

---

> ### Author Response · Authors · 2024-12-02
>
> Dear Reviewer tEeX,
>
> Thank you for reviewing our paper! With the discussion period ending in one day, we wanted to follow up as we haven’t heard from you. We would appreciate your feedback to ensure all concerns are addressed. If resolved, we kindly ask for your consideration in revising the scores.

---

> > ### Comment · Reviewer_tEeX · 2024-12-02
> > **response**
> >
> > I would like to thank the authors for the responses with lots of details. The responses answer most of my questions satisfyingly.
> >
> > The remaining problem is the interpretability of the approach (related to Q2 and A2).
> >
> > > Note that f_LM is a black-box and our goal is to learn mapping f_CBL from black-box representations to human-understandable concepts that are task-relevant.
> >
> > To me, the concept bottleneck has nothing to do with LLMs at all, because it treats an LLM as a blackbox and there's no mechanism for it to look into the blackbox. We can essentially replace the LLM with anything else. Thus, the CBL can only "interpret" representations fed to the output layer, not the LLM or any blackbox used in the framework. If this is correct, I believe the claim about interpretability of the paper is unjustified.

---

> ### Author Response · Authors · 2024-12-02
> **Further clarification regarding our claim of building interpretable LLMs**
>
> Dear Reviewer tEeX,
>
> Thank you for your response!
> We would like to clarify a potential misunderstanding regarding the claim of building interpretable LLMs in our paper. In Step 4 of Section 3.1, $f_{LM}$ refers to the “pretrained” LM, which generates hidden representations but is not directly applicable to the classification task. **Our goal is not to explain the “original pretrained” LM but to transform it into a more interpretable CB-LLM that provides transparency in its predictions.**
>
> By fine-tuning the entire CB-LLM including $f_{LM}$, the resulting model should be regarded as a new, distinct model. The CB-LLM is indeed inherently interpretable (as it now becomes an interpretable model by design with the added concept bottleneck layer; the prediction is transparent which is a linear combination of the concepts), which supports our claim of building interpretable LLMs in the paper.
>
> Moreover, in both computer vision and natural language processing, prior works [1,2,3,4,5,6] on CBM consistently claim to be interpretable while including a backbone model that functions as a black box. Achieving full interpretability for every neuron in an LLM remains an open challenge and is beyond the current state of research. We believe this is a broader topic that requires further exploration and should not overshadow the contributions and value of this work.
>
> We see our research as a step toward the ultimate goal of fully transparent LLMs. By leveraging CBMs, we aim to inspire future studies that advance this vision. We sincerely hope the above perspective addresses your concerns.
>
> [1] Koh etal., Concept Bottleneck Models, ICML 2020
>
> [2] Yuksekgonul etal., Post-hoc Concept Bottleneck Models, ICLR 2023
>
> [3] Oikarinen etal., Label-Free Concept Bottleneck Models, ICLR 2023
>
> [4] Yang etal., Language in a Bottle: Language Model Guided Concept Bottlenecks for Interpretable Image Classification, CVPR 2023
>
> [5] Ludan etal., Interpretable-by-Design Text Understanding with Iteratively Generated Concept Bottleneck, arXiv 2023
>
> [6] Tan etal., Interpreting Pretrained Language Models via Concept Bottlenecks, arXiv 2023

---

> > ### Author Response · Authors · 2024-12-03
> >
> > Dear Reviewer tEeX,
> >
> > Thank you for raising the remaining concerns! Since reviewers will no longer be able to post after a few hours, we wanted to kindly follow up to confirm if our clarification of our claim and goal has addressed your concerns. We would sincerely appreciate your consideration in reevaluating the score.

---

> > > ### Comment · Reviewer_tEeX · 2024-12-03
> > > **response**
> > >
> > > I would like to thank the authors to clarify the interpretability point. After carefully considering the whole thread, I raised my final score.

---

> > > > ### Author Response · Authors · 2024-12-03
> > > >
> > > > Dear Reviewer tEeX,
> > > >
> > > > Thank you for acknowledging the interpretability of CB-LLM. We sincerely appreciate your efforts in reviewing our paper and providing valuable feedback to help us improve the draft.

---

### Official Review · Reviewer_we68 · 2024-11-03

**Soundness:** 3
**Presentation:** 3
**Contribution:** 3
**Rating:** 6
**Confidence:** 3

**Summary:**

The paper presents CB-LLM, designed to provide inherent interpretability for both text classification and text generation tasks. In text classification, the model performs comparably to traditional black-box models while offering interpretable reasoning. For text generation, the model introduces interpretable neurons that enable concept detection and controllable text creation, presenting potential applications such as chatbot toxicity moderation.

**Strengths:**

- The paper presents a novel method for efficient interpretation in text classification tasks and proposes the first method for interpretation in text generation tasks.
- Both methods have somewhat practical applications for real-world tasks.
- The paper is easy to follow and includes rich figures and examples.

**Weaknesses:**

1. There is limited discussion about the generalizability of interpretable layers across various domains or tasks, specifically regarding whether concepts learned in classification or generation from one dataset can be reused in another. This lack of focus on transfer learning limits the model's broader applicability to related but unseen tasks. Moreover, the method is only tested on LLaMA-3-8B, so it is also important to investigate whether it consistently works across different model families and sizes.
1. It remains uncertain if the generated concepts consistently align with human intuition across all datasets.
1. The reliance on ChatGPT and sentence embedding models raises concerns about performance variability. Given their close performance levels, choosing different models or conducting multiple experiments could help mitigate this variability.

**Questions:**

1. With reference to Section 3.2, I am uncertain about the precise number of trained models: RoBERTa, GPT-2, and LLaMA-3. Could the authors succinctly clarify the training process and the employed LM, LLM, and sentence embedder?
2. In Figure 1, should the value prior to correction be **-** 0.1?

---

> ### Author Response · Authors · 2024-11-21
> **Rebuttal (1/3)**
>
> Dear Reviewer we68,
>
> Thank you for your positive feedback! We would like to address your concerns on the Weaknesses section and also provide answers to your questions.
>
> > **Q1:** There is limited discussion about the generalizability of interpretable layers across various domains or tasks, specifically regarding whether concepts learned in classification or generation from one dataset can be reused in another. This lack of focus on transfer learning limits the model's broader applicability to related but unseen tasks. Moreover, the method is only tested on LLaMA-3-8B, so it is also important to investigate whether it consistently works across different model families and sizes.
>
> **A1:** Thank you for the suggestion! Firstly, to address your concerns about generalizability, we have conducted additional experiments on Out-of-distribution (OOD) tests for both classification and generation tasks. Specifically, we evaluated models trained on SST-2 using IMDb as the test dataset, and models trained on Yelp Reviews using Amazon Reviews. The results, presented in below Tables A and B, demonstrate that CB-LLMs exhibit strong generalization to unseen tasks in both classification and generation settings, sometimes even outperforming the black-box models.
>
> Second, following your suggestion, we conducted additional experiments to study the impact of different model families and sizes. Specifically, we trained CB-LLM (classification) using Gemma2-2B as the new backbone and CB-LLM (generation) using Llama2-13B and Mistral-7B as new backbones. We show the results in below Tables C and D, and it can be seen that CB-LLMs achieve comparable performance to their black-box counterparts regardless of the backbone used, highlighting the robustness and adaptability of our pipeline across diverse models.
>
> **Table A:** OOD test for classification accuracy of CB-LLM (classification)
> | Accuracy | Train on: SST2  Evaluate on: IMDb | Train on: Yelp polarity Evaluate on: Amazon polarity |
> |---|---|---|
> | CBLLM w/ ACC (Ours) | 0.9117 | 0.9489 |
> | Roberta black box (Baseline) | 0.9066 | 0.9422 |
>
> **Table B:** OOD test for concept detection of CB-LLM (generation)
> | Accuracy | Train on: SST2  Evaluate on: IMDb | Train on: Yelp polarity Evaluate on: Amazon polarity |
> |---|---|---|
> | CBLLM (Ours) | 0.923 | 0.954 |
> | Llama3 black box (Baseline) | 0.880 | 0.955 |
>
> **Table C:** The accuracy of CBLLM (classification) using Gemma2-2b as the backbone
> | Backbone (Gemma2-2b) | SST2 | AGNews |
> |---|---|---|
> | CBLLM w/ ACC (Ours) | 0.9594 | 0.9471 |
> | Gemma2 black box (Baseline) | 0.9610 | 0.9538 |
>
> **Table D:** The accuracy, steerability, and perplexity of CBLLM (generation) using Llama2-13b and Mistral-7b as the backbone
> | Backbone | Metric | SST2 | AGNews |
> |---|---|---|---|
> | CBLLM (Llama2-13b) (Ours) | accuracy↑ | 0.9649 | 0.9444 |
> |  | steerability↑ | 0.86 | 0.90 |
> |  | perplexity↓ | 78.86 | 30.61 |
> | CBLLM (Mistral-7b) (Ours) | accuracy↑ | 0.9500 | 0.9428 |
> |  | steerability↑ | 0.82 | 0.62 |
> |  | perplexity↓ | 55.25 | 23.22 |
> | Llama2-13b blackbox (baseline) | accuracy↑ | 0.9676 | 0.9525 |
> |  | steerability↑ | No | No |
> |  | perplexity↓ | 31.29 | 22.99 |
> | Mistral blackbox (baseline) | accuracy↑ | 0.9594 | 0.9493 |
> |  | steerability↑ | No | No |
> |  | perplexity↓ | 32.70 | 21.78 |
>
> > **Q2:** It remains uncertain if the generated concepts consistently align with human intuition across all datasets.
>
> **A2:** Thank you for the comment! We would like to clarify that in Sec 3.2, an extensive human study is conducted across four benchmarks to verify that the explanations provided by CB-LLM align with human intuition. Please see Table 3 and Table 4 in the draft, which shows that our models are more interpretable and provide better explanations than prior work. The visualizations of the interpretable neurons and the generated explanations are also provided in Appendix A.6 and A.7, allowing readers to further evaluate the interpretability of CB-LLM.

---

> ### Author Response · Authors · 2024-11-21
> **Rebuttal (2/3)**
>
> > **Q3:** The reliance on ChatGPT and sentence embedding models raises concerns about performance variability. Given their close performance levels, choosing different models or conducting multiple experiments could help mitigate this variability.
>
> **A3:** Thank you for the suggestion! Following your suggestion, we first conducted additional experiments to address the variability in the concept generation process (using ChatGPT) and automatic concept scoring (using sentence embedding models). For concept generation, as ChatGPT’s responses are non-deterministic, we generated a new concept set and repeated the training process. We report our results in below Table E, which show that there is no significant difference in performance, indicating that the variability in concept generation is minimal.
>
> Next, for automatic concept scoring, we conducted additional experiments by replacing the MPNet model used in our paper with an alternative sentence embedding model, SimCSE [1]. We report the results in below Table F, which show that while MPNet achieves slightly higher accuracy than SimCSE, the performance gap is minor. This suggests that even using older sentence embedding models like SimCSE could still deliver high performance.
>
> **Table E:** Training CB-LLM (classification) with different concept set from ChatGPT
> | Accuracy | SST2 | AGnews |
> |---|---|---|
> | Regenerate concept set with the same size: |  |  |
> | CB-LLM w/ ACC | 0.9434 | 0.9420 |
> | Reported in our paper: |  |  |
> | CB-LLM w/ ACC | 0.9407 | 0.9453 |
>
> **Table F:** CBLLM (classification) using SimCSE as sentence embedding model
> |  | SST2 | Yelp | AGNews | DBpedia |
> |---|---|---|---|---|
> | SimCSE ACS |  |  |  |  |
> | CBLLM | 0.8852 | 0.9234 | 0.8901 | 0.9754 |
> | CBLLM w/ ACC | 0.9330 | 0.9789 | 0.9439 | 0.9922 |
> | MPNet ACS |  |  |  |  |
> | CBLLM | 0.9012 | 0.9312 | 0.9009 | 0.9831 |
> | CBLLM w/ ACC | **0.9407** | **0.9806** | **0.9453** | **0.9928** |
>
> [1] Gao etal., SimCSE: Simple Contrastive Learning of Sentence Embeddings, EMNLP 2021
>
> > **Q4:** With reference to Section 3.2, I am uncertain about the precise number of trained models: RoBERTa, GPT-2, and LLaMA-3. Could the authors succinctly clarify the training process and the employed LM, LLM, and sentence embedder?
>
> **A4:** Section 3.2 is the experiment for CB-LLM in the classification setting. We describe the model setting for CB-LLM and the baseline (TBM&C3M) as follows:
>
> **CB-LLM:**
> * Concept generation: ChatGPT
> * Pseudo concept labeling (Automatic Concept Scoring): MPNet (sentence embedding model)
> * Training: Roberta and GPT2
>
> **Baseline (TBM&C3M):**
> * Concept generation: ChatGPT
> * Pseudo concept labeling: Llama3-Instruct
> * Training: Roberta and GPT2
>
> Concept generation is performed using ChatGPT for both CB-LLM and the baseline. However, the labeling process differs: we propose Automatic Concept Scoring (Step 2), which leverages a sentence embedding model, providing a significantly more cost-effective and faster solution, achieving at least a 1000x speed-up (please see Appendix A.3, Table 7). In contrast, the baseline relies on LLMs for labeling.
>
> For training, only RoBERTa and GPT-2 are used during the training phase. The experiments are conducted across four datasets: SST-2, Yelp Polarity, AGNews, and DBpedia. As a result, we fine-tune four RoBERTa models and four GPT-2 models, one for each dataset.
>
> > **Q5:** In Figure 1, should the value prior to correction be - 0.1?
>
> **A5:** Thank you for the question! We note that the value of 0.1 before correction is not a typo. It is a wrong activation for the concept “Reliable Cleaning,” which is a positive concept. In Figure 1, the input sample provided is “Worst company ever!!!! No customer service. If you call on Sunday you're out of luck and they don't care!!”, which is a negative example from the Yelp review dataset. In this case, our Automatic Concept Correction in Step 3 will adjust the activation by setting it to zero (shown in green color), as a positive concept should not be activated for a negative sample.

---

> ### Author Response · Authors · 2024-11-21
> **Rebuttal (3/3)**
>
> ### **In summary,**
> * For **Q1**, we provide additional OOD tests for CB-LLM and train CB-LLM across a wide range of backbones, which verifies the strong generalization ability of our pipeline.
> * For **Q2**, we clarify that we have conducted extensive human studies in Sec 3.2 to show the generated concepts aligned well with human intuitions and provided visualization in Appendix A.6 and A.7 for readers to verify.
> * For **Q3**, we regenerate the concept set and do the training again to mitigate the concern of variability. We also use SimCSE to replace MPNet to show that our pipeline works for different sentence embedding models.
> * For **Q4**, we clarify the details of our setting in Section 3.2.
> * For **Q5**, we clarify that Figure 1 is correct.
>
> Please let us know if you still have any remaining concerns or questions and we would be happy to discuss further!

---

> > ### Comment · Reviewer_we68 · 2024-11-21
> > **Response to Rebuttal**
> >
> > Thanks for your detailed experiments and explanations. I remain the original score as a positive feedback.

---

> ### Author Response · Authors · 2024-11-23
>
> Dear Reviewer we68,
>
> Thank you for your positive feedback! We have added all the experiments during the rebuttal to Appendix C, which is structured as follows:
>
> **Appendix C.1:** CB-LLM (classification and generation) trained with different backbones
>
> **Appendix C.2:** OOD test of CB-LLM (classification and generation)
>
> **Appendix C.3:** The accuracy of CB-LLM (classification) using SimCSE Automatic Concept Scoring (ACS)
>
> **Appendix C.4:** Additional human study for CB-LLM (generation)
>
> **Appendix C.5:** Results of CB-LLM (generation) without using class labels
>
> **Appendix C.6:** The steerability of CB-LLM compared with the results without intervention
>
> **Appendix C.7:** The agreement coefficient of human study for classification case (Task1 and Task2 in Section 3.2)

---

### Official Review · Reviewer_mhCK · 2024-11-04

**Soundness:** 3
**Presentation:** 2
**Contribution:** 3
**Rating:** 5
**Confidence:** 3

**Summary:**

This paper introduces the Concept Bottleneck Large Language Model (CB-LLM), a novel large language model (LLM) that offers clear and accurate explanations through built-in interpretability and scalability, representing a breakthrough compared to traditional black-box LLMs. CB-LLM not only narrows the performance gap with black-box models in text classification tasks but also demonstrates how to leverage interpretable neurons for concept detection and guided text generation in text generation tasks. The authors conducted experiments across multiple datasets and metrics, showing that CB-LLM achieves considerable performance while maintaining inherent interpretability compared to black-box models. Additionally, the authors point out that CB-LLM not only possesses inherent interpretability during text generation but also provides greater controllability over the generated content. This feature gives CB-LLM potential application value in various fields requiring controllable content generation.

**Strengths:**

1. CB-LLM demonstrates outstanding performance in text classification tasks. After the introduction of Automatic Concept Correction (ACC), the model surpasses traditional black-box models on several datasets, showcasing its strong potential.
2. During the training process for text classification, CB-LLM effectively removes negative activation values using the ReLU activation function, significantly reducing semantic ambiguity during training and enhancing model stability.
3. The authors provide evidence through human evaluation that CB-LLM received positive ratings for both Activation Faithfulness and Contribution Faithfulness, indicating that the model's explanations align more closely with human understanding and expectations, thereby enhancing the model's transparency and trustworthiness.

**Weaknesses:**

1. In the CB-LLM for text classification , the introduction of sparsity constraints during training is not adequately explained. If this constraint aims to enhance the model's interpretability, the authors should provide more analysis and experimental evidence to support this claim.
2. The authors used only RoBERTa and GPT-2 as backbone models in their experiments. Given that the main goal of the paper is to improve the internal interpretability of large language models (LLMs) in text classification, it would be beneficial to conduct related experiments using larger models, rather than relying solely on LLaMA for labeling.
3. Although CB-LLM applies the concept bottleneck to text generation tasks, there is a lack of sufficient experimental validation regarding its internal interpretability when generating text. In section 4.2, the “concept detection” experiment is conducted only on a text classification dataset, which does not effectively demonstrate whether CB-LLM's internal interpretability in text generation is faithful and valid.
4. Some statements in the paper may need to be revised for clarity and to avoid ambiguity. For example, the variable “y” in Equation 2 lacks sufficient contextual explanation. Additionally, both Table 2 and Table 6 mention “RoBERTa-base fine-tuned (black-box),” but the values differ. The “RoBERTa-base fine-tuned (black-box)” in Table 6 should be replaced with “GPT-2” to eliminate confusion.

**Questions:**

1. I find the illustration in Figure 3 somewhat confusing. Module 1 is represented in green text; does this mean that all green parts belong to Module 1? Similarly, do the blue parts represent Module 2? Due to the positioning of the text, I have difficulty distinguishing between the two modules. If my understanding is correct, I recommend adding some identifiers to clarify this distinction.
2. I found it difficult to understand the concept of "concept bottleneck", which is widely used in the field of computer vision. I hope that a brief description of the concept bottleneck can be included in the related work section, as this would help readers from unrelated fields to quickly understand the paper.

---

> ### Author Response · Authors · 2024-11-21
> **Rebuttal (1/3)**
>
> Dear Reviewer mhCK,
>
> Thank you for your valuable feedback and detailed comments! We would like to address your concerns on the Weaknesses section and also provide answers to your questions.
>
> > **Q1:** In the CB-LLM for text classification, the introduction of sparsity constraints during training is not adequately explained. If this constraint aims to enhance the model's interpretability, the authors should provide more analysis and experimental evidence to support this claim.
>
> **A1:** Thanks for your comments! The idea of adding sparsity constraints is first proposed in the prior CBM works ([1, see Section 3.3], [2, see page 4 top], [3, see Section 3.3]) in the vision domains to encourage only a few key concepts in the concept-bottleneck layer to contribute to the prediction. It helps interpretability because the predictions are associated with only a few important concepts instead of several tens or hundreds of concepts, which could make it complicated for humans to follow and understand.
>
> Following your suggestions, we provide additional analysis on sparsity through a human study conducted on MTurk, as shown in Table A. Workers show a preference (7-9% higher) for the models with the sparsity constraints in both w/ ACC and w/o ACC settings. This finding verifies the effectiveness of the sparsity constraint in enhancing interpretability.
>
> **Table A:** The effect of the sparsity constraint for CB-LLM
> | CB-LLM w/ ACC |  |  |
> |---|---|---|
> | w/ sparsity is better | Equally good | w/o sparsity is better |
> | **38%** | 33% | 29% |
> | **CB-LLM** |  |  |
> | w/ sparsity is better | Equally good | w/o sparsity is better |
> | **39%** | 29% | 32% |
>
> [1] Oikarinen etal., Label-Free Concept Bottleneck Models, ICLR 2023
>
> [2] Yuksekgonul etal., Post-hoc Concept Bottleneck Models, ICLR 2023
>
> [3] Srivastava etal., VLG-CBM: Training Concept Bottleneck Models with Vision-Language Guidance, NeurIPS 2024
>
>
> > **Q2:** The authors used only RoBERTa and GPT-2 as backbone models in their experiments. Given that the main goal of the paper is to improve the internal interpretability of large language models (LLMs) in text classification, it would be beneficial to conduct related experiments using larger models, rather than relying solely on LLaMA for labeling.
>
> **A2:** Following your suggestions, we have conducted additional experiments using larger models for both classification and generation tasks, with results presented in the below Tables B and C. In the classification setting, we utilize the larger GEMMA 2 model than RoBERTa to construct the CB-LLM. The accuracy of our CB-LLM remains comparable to the black-box baseline, demonstrating that our method scales effectively to larger models. For the generation setting, we employ a larger model (LLaMA 2-13B) as the backbone model than the 8B models in the manuscript. The new results confirm that our pipeline continues to perform well, achieving strong outcomes in accurate concept detection, high steerability, and low perplexity.
>
> **Table B:** The accuracy of CB-LLM (classification) using Gemma2-2b as the backbone
> | Backbone (Gemma2-2b) | SST2 | AGNews |
> |---|---|---|
> | CB-LLM w/ ACC (Ours) | 0.9594 | 0.9471 |
> | Gemma2-2b black box (Baseline) | 0.9610 | 0.9538 |
>
> **Table C:** The accuracy, steerability, and perplexity of CB-LLM (generation) using Llama2-13b as the backbone
> | Backbone (Llama2-13b) | Metric | SST2 | AGNews |
> |---|---|---|---|
> | CBLLM (Ours) | accuracy↑ | 0.9649 | 0.9444 |
> |  | steerability↑ | 0.86 | 0.90 |
> |  | perplexity↓ | 78.86 | 30.61 |
> | Llama2-13b (Baseline) | accuracy↑ | 0.9676 | 0.9525 |
> |  | steerability↑ | No | No |
> |  | perplexity↓ | 31.29 | 22.99 |

---

> ### Author Response · Authors · 2024-11-21
> **Rebuttal (2/3)**
>
> > **Q3:** Although CB-LLM applies the concept bottleneck to text generation tasks, there is a lack of sufficient experimental validation regarding its internal interpretability when generating text. In section 4.2, the “concept detection” experiment is conducted only on a text classification dataset, which does not effectively demonstrate whether CB-LLM's internal interpretability in text generation is faithful and valid.
>
> **A3:** Following your suggestion, we have conducted an additional human study experiment to evaluate whether the text generation of CB-LLM faithfully reflects neuron activations. In our experiment, workers were provided with a generated sentence from CB-LLM and the corresponding neuron concept in the CBL which has the highest activation, and their task is to evaluate whether the generated sentence matches the specified concept. We evaluate 100 generated sentences for each dataset and report in Table D. The results show that there is an average match rate of 90%. This suggests that when CB-LLM generates concept-related sentences, the corresponding neuron is indeed highly active, thereby demonstrating the faithfulness of the generation process.
>
> Additionally, for concept detection in Section 4.2, we performed another experiment using out-of-distribution (OOD) tests to validate that the concept detection mechanism works effectively across diverse inputs. The results, shown in Table E, reveal that CB-LLM achieves a detection accuracy comparable to (or even better than) that of the black-box Llama3 fine-tuned for concept detection.
>
> **Table D:** CB-LLM generation human study. Survey question: “Does the generated sentence match the neuron activations in CBL?”
> | Match rate | SST2 | Yelp | AGNews | DBpedia | Avg |
> |---|---|---|---|---|---|
> | CB-LLM (generation) | 0.8867 | 0.9133 | 0.9133 | 0.8867 | 0.9 |
>
> **Table E:** OOD test for concept detection of CB-LLM (generation)
> | Accuracy | Train on: SST2,  Evaluate on: IMDb | Train on: Yelp polarity, Evaluate on: Amazon polarity |
> |---|---|---|
> | CB-LLM (Ours) | 0.923 | 0.954 |
> | Llama3 black box (Baseline) | 0.880 | 0.955 |
>
> > **Q4:** Some statements in the paper may need to be revised for clarity and to avoid ambiguity. For example, the variable “y” in Equation 2 lacks sufficient contextual explanation. Additionally, both Table 2 and Table 6 mention “RoBERTa-base fine-tuned (black-box),” but the values differ. The “RoBERTa-base fine-tuned (black-box)” in Table 6 should be replaced with “GPT-2” to eliminate confusion.
>
> **A4:** Thanks for the suggestion! In the original draft, we stated before introducing Equation 2 that “Let y be the class label of x,” where y represents the class label in the dataset. We would like to make this point more clear here: For example, in SST-2, there are two classes: negative and positive. Specifically, y=0 indicates that the sample x is labeled as negative, while y=1 means the sample x is labeled as positive.
>
> Regarding the baseline in Table 6, thank you for pointing out the typo. The model should be GPT-2 black-box instead of RoBERTa black-box. We have revised our paper accordingly in Appendix A.2 page 15.
>
> > **Q5:** I find the illustration in Figure 3 somewhat confusing. Module 1 is represented in green text; does this mean that all green parts belong to Module 1? Similarly, do the blue parts represent Module 2? Due to the positioning of the text, I have difficulty distinguishing between the two modules. If my understanding is correct, I recommend adding some identifiers to clarify this distinction.
>
> **A5:** Yes, all the components in module 1 are colored in green and all the blue parts are the adversarial training module. Thanks for your suggestion, we have revised Figure 3 in the draft to make it clearer.
>
> > **Q6:** I found it difficult to understand the concept of "concept bottleneck", which is widely used in the field of computer vision. I hope that a brief description of the concept bottleneck can be included in the related work section, as this would help readers from unrelated fields to quickly understand the paper.
>
> **A6:** Thank you for the suggestion, we have added more description about the Concept Bottleneck Model in the Related work section. Please see the blue text in our paper at the beginning of Section 2.

---

> ### Author Response · Authors · 2024-11-21
> **Rebuttal (3/3)**
>
> ### **In summary,**
> * For **Q1**, we provide additional human studies and show that sparsity can enhance interpretability.
> * For **Q2**, we do additional training to verify that CB-LLM works well with larger backbones for both classification and generation.
> * For **Q3**, We conduct human studies to verify CB-LLM's faithfulness and OOD tests to assess its generalization ability.
> * For **Q4**, we clarify what is “y” in Equation 2 and fix the typo in Table 6.
> * For **Q5**, we revise Figure 3 to improve the clarity.
> * For **Q6**, we add more background information about CBM at the beginning of Section 2.
>
> Please let us know if you still have any remaining concerns or questions and we would be happy to discuss further!

---

> ### Author Response · Authors · 2024-11-24
>
> Dear Reviewer mhCK,
>
> As the discussion will end in a few days, please let us know if we have addressed your concerns. We would also be happy to address any further questions you may have. We greatly appreciate your feedback and look forward to hearing from you!

---

> ### Author Response · Authors · 2024-11-27
> **Request rebuttal feedback: we would love to hear from you**
>
> Dear Reviewer mhCK,
>
> We would like to kindly follow up to request your valuable feedback. We truly appreciate your thoughtful comments and have worked diligently to address your major concerns as follows:
>
> 1. We performed extensive additional training and testing of CB-LLM to ensure that our pipeline generalizes effectively to a wide range of models and OOD data **(please see Q2 and Q3)**.
> 2. We conducted comprehensive human studies to assess the impact of sparsity and verify the faithfulness of CB-LLM's generated outputs **(please see Q1 and Q3)**.
> 3. Based on your suggestions for writing, we have included the requested updates in the blue-highlighted text at the beginning of Section 2 and added a new Figure 3.
>
> Thank you again for your constructive feedback, which has been instrumental in enhancing our work. If we have addressed all your concerns, we kindly request your consideration in revising the scores. If you still have any additional comments, we would be happy to discuss them further.

---

> ### Author Response · Authors · 2024-12-02
>
> Dear Reviewer mhCK,
>
> Thank you for reviewing our paper! With the discussion period ending in one day, we wanted to follow up as we haven’t heard from you. We would appreciate your feedback to ensure all concerns are addressed. If resolved, we kindly ask for your consideration in revising the scores.

---

> ### Author Response · Authors · 2024-12-03
>
> Dear Reviewer mhCK,
>
> Thank you for reviewing our paper! Since reviewers will no longer be able to post after a few hours, we wanted to follow up as we haven’t heard from you. We would appreciate your feedback to ensure all concerns are addressed. If resolved, we kindly ask for your consideration in revising the scores.

---

### Meta-Review · Area_Chair_sagZ · 2024-12-24

**Metareview:**

`This paper applies the interpretability technique presented by concept bottleneck methods to text classification and generation tasks. The method allows for a mapping between concepts generated by GPT and the label set for classification tasks. A similar approach is used for text generation.

**Strengths:** The paper is well motivated and extends an existing method from other domains to text. Most importantly, it provides a method for text generation, which is arguably less studied in interpretability literature than text classification.

**Weaknesses:** While the method shows some improvements in interpretability, it results in a much worse language model, since perplexity is hurt, often by a lot. Use of many terms is non-standard - e.g. “activation of neurons”, “steerability”.

**Reason for acceptance (poster)**: The paper provides an application of the concept bottleneck method to language models and has shown empirical results on a variety of settings thanks to the reviewer feedback.

**Additional Comments On Reviewer Discussion:**

Based on the reviewer feedback, the authors seem to have provided more empirical evidence for sparsity, experiments with larger language models - Llama, Gemma, human experiments for interpretability, answered questions about why the method is interpretable, and presented results for accuracy, perplexity and steerability of their methods and the baselines. The reviewer discussion might have contributed to improving the overall contribution of the paper. The authors have provided extremely detailed comments explaining their changes.

---

### Decision · Program_Chairs · 2025-01-22

Accept (Poster)